# Mastering Robot Manipulation with Multimodal Prompts through Pretraining and Multi-task Fine-tuning

## Abstract

Prompt-based learning has been demonstrated as a compelling paradigm contributing to large language models' tremendous success (LLMs). Inspired by their success in language tasks, existing research has leveraged LLMs in embodied instruction following and task planning. However, not much attention has been paid to embodied tasks with multimodal prompts, combining vision signals with text descriptions. This type of task poses a major challenge to robots' capability to understand the interconnection and complementarity between vision and language signals. In this work, we introduce an effective framework that learns a policy to perform robot manipulation with multimodal prompts from multi-task expert trajectories. Our methods consist of a two-stage training pipeline that performs inverse dynamics pretraining and multi-task finetuning. To facilitate multimodal understanding, we design our multimodal prompt encoder by augmenting a pretrained LM with a residual connection to the visual input and model the dependencies among action dimensions. Empirically, we evaluate the efficacy of our method on the VIMA-BENCH (Jiang et al., 2023) and establish a new state-of-the-art (10% improvement in success rate). Moreover, we demonstrate that our model exhibits remarkable in-context learning ability.

## 1 Introduction

The unprecedented advancement of large language models (LLM) (Brown et al., 2020; OpenAI, 2023; Chowdhery et al., 2022; Anil et al., 2023; Chung et al., 2022; Touvron et al., 2023) has stimulated rapid development of building instruction-following agents (Lynch & Sermanet, 2020; Ahn et al., 2022; Driess et al., 2023; Guhur et al., 2023; Huang et al., 2022a). By leveraging LLM's remarkable zero-shot generalizability, various research initiatives Ahn et al. (2022); Huang et al. (2022a;b) have developed powerful action planners to parse language instructions into a sequence of sub-goals. A prominent example is the SayCan (Ahn et al., 2022), which employs PALM (Chowdhery et al., 2022) to transform abstract task descriptions into actionable step-by-step plans.

However, relying solely on language instructions can be inefficient for describing intricate task details. For instance, directing a household robot to tidy out a living room is more straightforward with a combination of language and visual cues than using language alone. Also, when learning new tasks, words simply cannot convey as much information as video demonstrations (Dasari & Gupta, 2021). In addition, human communication is inherently multimodal, often combining speech with expressive gestures and demonstrations (Drijvers & Holler, 2023). Therefore, we are motivated to enhance a robot's comprehension of multimodal task prompts that interleave text and images.

Training a robot to interpret multimodal prompts presents several challenges. The vision signals in the prompt may simply represent target objects, delineate a specific sub-goal, or offer in-context demonstrations. Here, the robot must first understand the underlying transition dynamics suggested by the multimodal prompts before tackling the overall task objective. This requires the robot to infer state transitions from language instructions, and also to be capable of inferring actions from image signals, i.e., inverse dynamic prediction. Furthermore, it is essential for the robot to pay attention to critical visual details, such as the orientation of an object shown in the image, as this can significantly influence its action prediction.

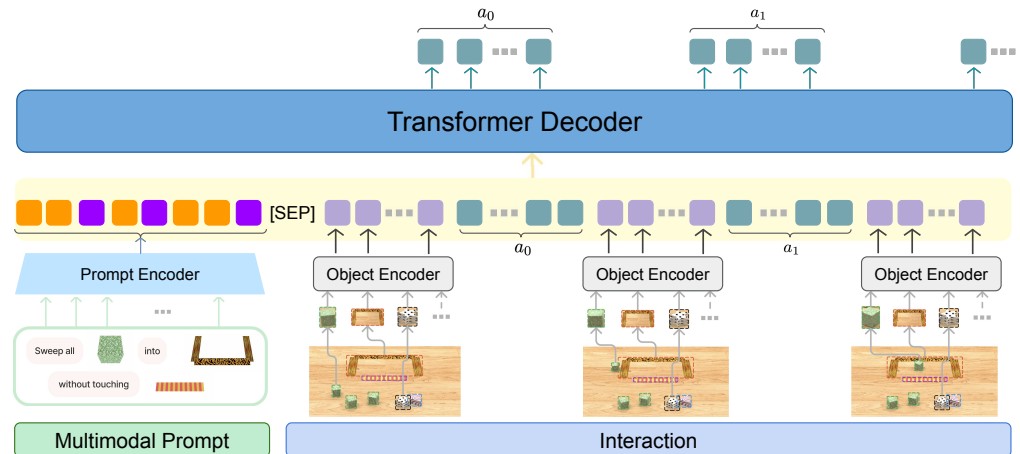

Figure 1: **Model Architecture**. Our model adopts a decoder-only architecture. The multimodal prompt embeddings are concatenated with history tokens. We model each action dimension as an individual token and predict them auto-regressively.

Matching object appearance with textual representation can be achieved by multi-task imitation learning on a diverse set of tasks (Jiang et al., 2023). However, this method falls short in teaching robots to predict inverse dynamics due to the masking of future observations in multi-task imitation learning. To overcome this challenge, we introduce a two-stage training pipeline: inverse dynamic pretraining and multi-task finetuning (FT). Our pretraining strategy first converts any robot trajectory into a motion-following task and then trains the robot to recover the action sequences given the observed image sequence. To capture fine-grained visual information, we design our multimodal prompt encoder by augmenting a pretrained LM with a residual connection (RC) adding from the input visual tokens to the encoded embeddings of the LM.

Figure 1 provides an overview of our model, which adopts a decoder-only architecture (Radford et al., 2018). Specifically, we model each action dimension as individual action token and predict them auto-regressively to capture dependencies among different dimensions. We dub our method as **M**ulti-modal **I**nverse **D**ynamics **A**gent**S** (MIDAS). Empirically, we evaluate our method on the VIMA-BENCH (Jiang et al., 2023) and establish a new state-of-the-art, outperforming VIMA by ∼10% on all 4 evaluation protocols of VIMA-BENCH. Furthermore, we showcase the our multi-task policy's superior in-context learning ability by modifying the original VIMA-BENCH. We emphasize this is novel, as simultaneously equipping a robot with multi-task and in-context learning abilities has not been extensively explored.

In summary, our contributions can be summarized as follows:

- Introduction of the two-stage MIDAS training framework, which establishes a new state-of-the-art on VIMA-BENCH (Jiang et al., 2023).
- An effective multimodal prompt encoder that can capture visual and textual details.
- Empirically highlighting the superior in-context learning ability of our multi-task policy.

## 2 PRELIMINARY

**Problem Definition** We consider the problem of learning a multimodal prompt-conditioned policy $\pi : \mathcal{P} \times \Omega \rightarrow \mathcal{A}$ that maps the multimodal prompt $q \in \mathcal{P}$ and the history trajectory $\omega_t = (o_0, a_0, o_1, \ldots, a_{t-1}, o_t) \in \Omega$ to the two-pose action primitive (Zeng et al., 2021) $a_t = (\mathcal{T}_{\text{initial}}, \mathcal{T}_{\text{target}}) \in \mathcal{A} \subseteq \mathcal{R}^{N_a}$, where $o_t \in \mathcal{O}$ denotes the visual observation at timestep $t$ and $N_a$ denotes the number of action dimensions.

$$\pi(q, \omega_t) = \pi(q, o_0, a_0, o_1, \ldots, a_{t-1}, o_t) \rightarrow a_t = (\mathcal{T}_{\text{initial}}, \mathcal{T}_{\text{target}}) \in \mathcal{A} \subseteq \mathcal{R}^{N_a} \quad (1)$$

The action space $\mathcal{A}$ consists of primitive motor skills like "pick and place" and "push". For the "pick and place" primitive, $\mathcal{T}_{\text{initial}}$ and $\mathcal{T}_{\text{target}}$ defines the space of pick pose and place pose, respectively.

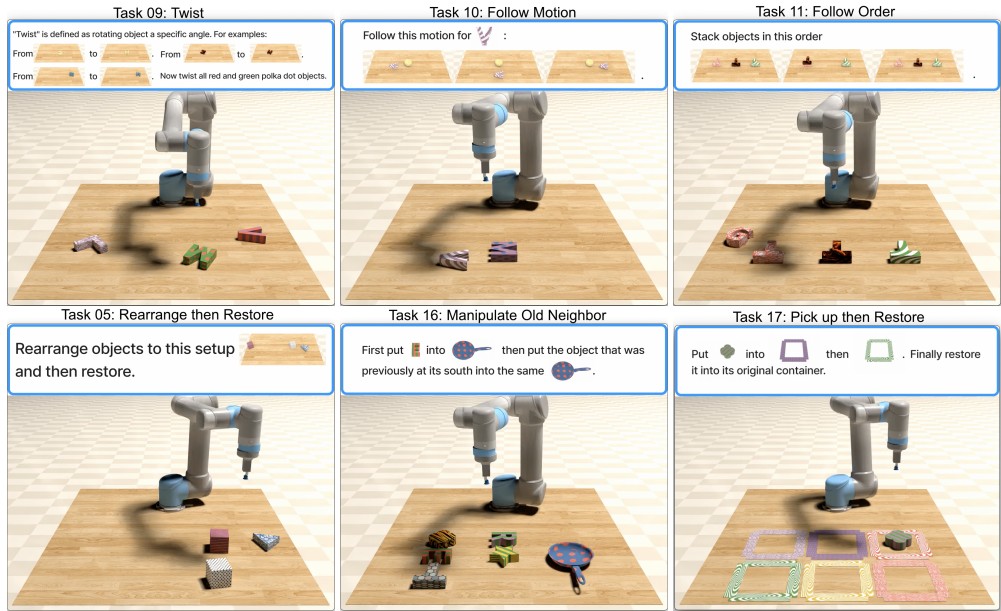

Figure 2: Task samples from the VIMA-BENCH. We refer readers to Appendix B of the VIMA paper (Jiang et al., 2023) for detailed task description.

For "push", they define the space of the starting and ending pose of push. The multimodal prompt describes the task goal by interleaving texts and images.

In this paper, we aim to learn a multi-task policy $\pi_\theta$ parameterized by $\theta$ from a dataset $\mathcal{D} = \{\zeta_1, \ldots, \zeta_N\}$ with $N$ expert demonstration. Each training sample $\zeta_i = (q^i, \omega^i)$ contains the expert trajectory $\omega^i = \left(o_0^i, a_0^i, o_1^i, \ldots, a_{T-1}^i, o_T^i\right)$ corresponding to the multimodal task prompt $q_i$.

**VIMA policy** Jiang et al. (2023) propose the VisuoMotor Attention (VIMA) agent to solve robot manipulation from multimodal prompts with a Transformer (Vaswani et al., 2017) Encoder-Decoder architecture. It encodes the multimodal prompts that interleave textual and visiual tokens with a pretrained LM by following the practice of Frozen (Tsimpoukelli et al., 2021). Its autoregressive action decoding is conditioned on the prompt embedding via cross attention layers that alternate with the causal self-attention. Instead of directly operating on the raw RGB images, VIMA adopts the object-centric representation by cropping objects from both prompt and observation images and forming them as a sequence of object tokens with pixel coordinate information as shown in Figure 3a. Notably, VIMA predicts each action dimension independently and trains its model via behavior cloning with the loss function for a trajectory with $T$ steps given by

$$L(\theta) = -\min_\theta \sum_{t=0}^{T-1} \log \pi_\theta(a_t|q, \omega_t) = -\min_\theta \sum_{t=0}^{T-1} \sum_{n=0}^{N_a-1} \log \pi_\theta(a_t^n|q, \omega_t). \qquad (2)$$

We build our policy upon the VIMA policy. However, we model the dependencies among different action dimensions (Giuliari et al., 2021; Vinyals et al., 2019) and decode each dimension autoregressively. We detail our motivation in Sec. 3.3 and demonstrate its empirical benefit in Sec. 4.

**VIMA-BENCH** (Jiang et al., 2023) is built on top of the Ravens (Zeng et al., 2021; Shridhar et al., 2023) simulator and contains 17 types of tabletop manipulation tasks. Figure 2 shows 6 representative tasks from the VIMA-BENCH. Each task type can instantiate thousands of individual task instances by combining various textures and objects. Specifically, each task instance defines a multimodal prompt that interleaves texts and images and the type of end-effector $\in$ {suction cup, spatula}. The suction cup corresponds to the primitive motor skill "pick and place" while spatula corresponds to "wipe". At each time step, the agent receives RGB images rendered from both frontal and top-down views and predicts the initial and target pose of its end effector.

VIMA-BENCH establishes a four-level protocol to evaluate progressively stronger generalization, ranging from placement generalization (L1), combinatorial generalization (L2), novel task general-

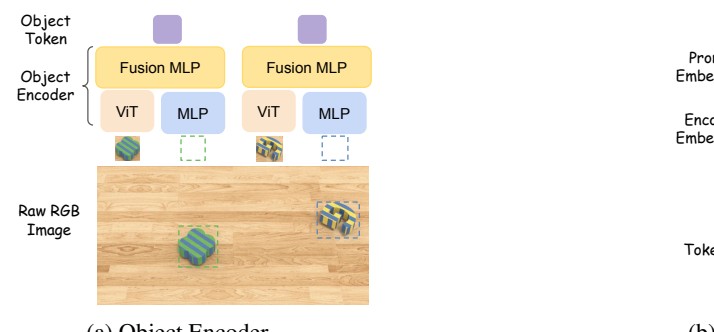
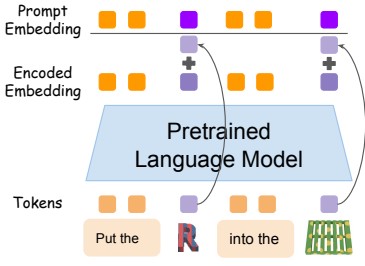

(a) Object Encoder

(b) Multimodal Prompt Encoder

Figure 3: (a) **Object Encoder** proposed in VIMA consists of a ViT (Dosovitskiy et al., 2020) that extracts visual embedding from cropped object images and a MLP that encodes bounding boxes. The two embeddings are concatenated before passing through a Fusion MLP to get the object tokens. (b) **Multimodal Prompt Encoder** adds a RC from the input object tokens to the pretrained LM output.

ization (L3) and novel task generalization (L4). Expert demonstration are provided for 13 tasks as the training data, with 50K trajectories per task. The other 4 tasks are included into the L4 task suite.

## 3 METHODS

We introduce our MIDAS framework that learns a multi-task policy to perform robot manipulation with multimodal prompt. We propose a two-stage training pipeline that includes inverse dynamic pretraining (Sec. 3.1) followed by multi-task FT. To capture fine-grained visual information, we design our multimodal prompt encoder by augmenting a pretrained LM with a residual connection to the input object token (Sec. 3.2). Moreover, we model each action dimension as an individual action token and autoregressively decodes each dimension (Sec. 3.3). Sec. 3.4 summarizes our training framwork, with an overview of our model architecture given in Figure 1.

### 3.1 PRETRAINING TASK: INVERSE DYNAMICS PREDICTION

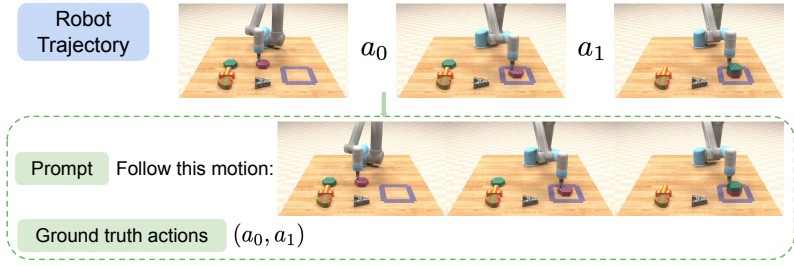

Figure 4: Given the any sequence of robot trajectory, we can always formulate a motion following task that requires the agent to replicate the demonstration trajectory.

As mentioned in Sec. 1, images in the prompt can depict object appearance, appearances, outline the sub-goals and success criteria of a task, or serve as in-context task demonstrations. To decipher this underlying task information and learn from in-context examples, a robot needs to understand the transition dynamics illustrated in a sequence of images. For instance, the robot should be able to infer the action sequence required to transition from its current state to the target goal state.

In other words, the agent needs proficiency in inverse dynamics prediction. Given a sequence of observations $(o_0, \ldots, o_T)$, the robot should learn to infer the corresponding action sequence $(a_0, \ldots, a_{T-1})$. However, the skill cannot be directly acquired by imitating multi-task trajectories, as future observations are often masked out when predicting actions with current observations.

To tackle the dilemma, we make a novel observation that every robot trajectory itself can be reformulated into a motion following task. As shown in Figure 4, given any sequence of robot trajectory

$\omega_T = (o_0, a_0, o_1, \ldots, a_{T-1}, o_T)$, we can always create a task with the prompt *Follow this motion: $o_0, \ldots, o_T$* and ground-truth actions $(a_0, \ldots, a_{T-1})$, leading to the following pretraining loss

$$L_{\text{pretrain}}(\theta) = - \min_\theta \sum_{t=0}^{T-1} \log \pi_\theta(a_t | \textit{Follow this motion: } o_0, \ldots, o_T; \omega_t) \tag{3}$$

## 3.2 MULTI-MODAL PROMPT ENCODING

To capture visual and textual information from the multimodal prompt, VIMA proposes to encode both the visual and language tokens in the prompt with a pretrained LM (T5-base) following the practice of Frozen (Tsimpoukelli et al., 2021). While LLM has demonstrated a tremendous success across various fields with superior generalizability (Li et al., 2022), our early experiments reveal that this encoding strategy often fails to capture some fine-grained visual information, e.g., the rotation angle of an object (Task 09, Figure 2). We hypothesize it is because the pretrained LM has never been trained on visual data.

To overcome this challenge, we propose to augment the pretrained LM by adding a residual connection (RC) from the input visual tokens to the encoded embeddings , as shown in Figure 3b. The intuition is that by directly adding the original visual tokens to the embeddings produced by the pretrained LM, we can retain more detailed visual information that might be lost during the encoding process. Our experiments in Sec. 4 validate this intuition, showing that the inclusion of the RC significantly improves performance across different tasks.

## 3.3 MODELING THE DEPENDENCY AMONG EACH ACTION DIMENSION

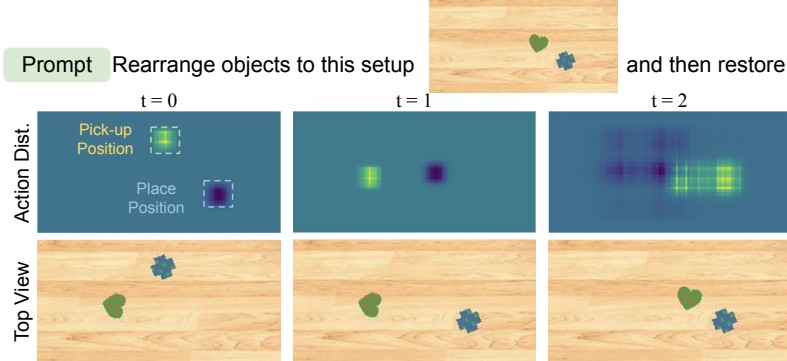

Figure 5: At $t = 2$, the robot can choose to move either the heart or the cross block. As the policy predicts each action dimension independently, different dimensions do not consistently manipulate the same object, resulting in a task failure.

Recall that the robot action is defined by the initial pose $\mathcal{T}_{\text{initial}}$ and target pose $\mathcal{T}_{\text{target}}$ of the end effector. Intuitively, $\mathcal{T}_{\text{target}}$ should depend on $\mathcal{T}_{\text{initial}}$. And thus independently predicting each action dimension can be problematic. Consider the example in Figure 5, the robot is tasked to first rearrange the objects to a desired arrangement and then restore them to original setup. When the robot begins to restore at $t = 2$, it has the option to move either the heart or the cross block. As the policy predicts each action dimension independently, the dimensions associated with the pick-up pose do not align consistently with one specific object. Consequently, the distribution of pick-up position assigns significant probability to both object locations. Similarly, the placement position distribution allocates probability to both objects' target positions. When sampling actions from this distribution, the robot may either miss picking up an object or misplace it, leading to a task failure.

Therefore, we opt to model the dependency among action dimensions by modeling each dimension as a single token and decode each token autoregressively as shown in Figure 1. And thus, the multi-task imitation loss function can be reformulated into

$$L_{\text{Imitation}}(\theta) = - \min_\theta \sum_{t=0}^{T-1} \left( \log \pi_\theta(a_t^0 | q, \omega_t) + \sum_{n=1}^{N_a-1} \log \pi_\theta(a_t^n | q, \omega_t, a_t^0, \ldots, a_t^{n-1}) \right). \tag{4}$$

Table 1: We compared our methods with baseline approaches on the VIMA-BENCH across all four evaluation levels. "Avg" represents the average success rate for all tasks within an evaluation level. To determine the success rate for each method, we sampled 200 episodes from every task. Due to limited space, we report the success rate for four representative tasks in this table. Full results can be found in Appendix A. Our methods significantly outperform baseline methods and establish a new state-of-the-art performance on the VIMA-BENCH.

| Method | L1 | | | | | L2 | | | | | L3 | | | | | L4 | |
|---|---|---|---|---|---|---|---|---|---|---|---|---|---|---|---|---|---|
| | Avg | T5 | T9 | T16 | T17 | Avg | T5 | T9 | T16 | T17 | Avg | T5 | T9 | T16 | T17 | Avg | T10 |
| Gato | 57.0 | 44.5 | 14.0 | 43.0 | 1.5 | 53.9 | 46.0 | 10.5 | 42.0 | 1.0 | 45.6 | 36.0 | 17.0 | 41.5 | 0.0 | 13.5 | 0.0 |
| Flamingo | 47.2 | 41.0 | 3.0 | 38.0 | 2.0 | 47.1 | 43.0 | 4.5 | 40.0 | 1.0 | 42.1 | 36.5 | 6.0 | 45.5 | 0.5 | 11.1 | 0.0 |
| GPT | 47.9 | 45.0 | 8.0 | 33.0 | 1.0 | 47.4 | 43.0 | 10.5 | 34.0 | 3.0 | 42.6 | 32.0 | 5.0 | 37.5 | 0.0 | 12.1 | 0.5 |
| VIMA | 87.2 | 65.0 | 13.5 | 88.0 | 77.0 | 87.0 | 61.0 | 12.5 | 87.5 | 77.5 | 84.0 | 63.0 | 12.0 | **58.5** | 78.0 | 49.6 | 0.0 |
| Gato OBJ | 87.5 | 62.0 | 17.0 | 92.5 | 80.5 | 87.5 | 62.5 | 16.0 | 91.5 | 80.0 | 84.4 | 65.5 | 15.5 | 46.5 | 87.5 | 49.6 | 0.0 |
| **Ours** | | | | | | | | | | | | | | | | | |
| w/o Pretrain | 91.6 | 88.0 | 20.5 | 93.0 | **98.0** | 91.8 | 87.0 | 23.5 | 92.0 | **98.0** | 88.3 | 90.0 | 20.5 | 50.5 | **99.5** | 49.1 | 0.0 |
| w/ Pretrain | **97.8** | **94.0** | **100** | **94.0** | 96.5 | **97.9** | **96.5** | **100** | **93.0** | 96.0 | **93.4** | **94.0** | **97.0** | 47.0 | **98.0** | **59.1** | **41.0** |

That is, the distribution for each action dimension should be conditioned on the other action dimensions that have already been decoded.

### 3.4 ALGORITHM SUMMARY

To this end, we have introduced our pretraining strategies and model design. To learn our multi-task policy $\pi_\theta$, we assume the access to a dataset $\mathcal{D} = \{\zeta_1, \ldots, \zeta_N\}$ with $N$ expert demonstration. First, we pretrain $\pi_\theta$ by minimizing $L_{\text{pretrain}}(\theta)$ over $N_{\text{pretrain}}$ iterations. Subsequently, we perform multi-task fine-tuning on $\mathcal{D}$ to minimize $L_{\text{Imitation}}(\theta)$. The pseudo-codes (Algorithm 1) and detailed hyper-parameters (HP) are available in Appendix B.

## 4 EXPERIMENTAL RESULTS

Our experiments focus on the VIMA-BENCH (Jiang et al., 2023), addressing two primary questions: 1) Does our model design and training pipeline enhance the zero-shot generalization of the learned model? 2) Can our model effectively utilize in-context examples to tackle novel tasks? To answer the first question, we evaluate our methods on the VIMA-BENCH (Sec. 4.1) and conduct extensive ablation studies (Sec. 4.2). To answer the second question, we modify the VIMA-BENCH by holding out more tasks with in-context examples from the training set to test our methods (Sec. 4.3).

### 4.1 STANDARD EVALUATION ON THE VIMA-BENCH

We compare our methods with various baselines from the original VIMA paper (Jiang et al., 2023) on the VIMA-BENCH. All baseline methods only conduct multi-task imitation learning without pretraining. We directly report results for Gato (Reed et al., 2022), Flamingo (Alayrac et al., 2022) and GPT (Radford et al., 2018) from the VIMA paper. Notably, these three methods directly operate on the raw image observation. In contrast, VIMA, Gato OBJ and our methods adopt an object-centric representation. The Gato OBJ policy is constructed by replacing the encoder-decoder architecture of the VIMA policy with a decoder-only architecture (Radford et al., 2018). And the difference between our policy and Gato OBJ is that we augments the pretrained LM with a RC and model each action dimension as an individual action token. As we do not focus on the visual understanding part of general robot control, we assume the access to the ground truth instance segmentation masks provided by the VIMA-BENCH for all methods with an object-centric representation. And the results of VIMA and Gato OBJ are reproduced by us.

Table 1 presents the evaluation results by following VIMA-BENCH's 4-level evaluation protocols. Due to the limited space, we only report the individual task success rates for representative tasks on which different methods exhibit a significant performance difference. Avg denotes the task success rate across all tasks from an evaluation level. Full evaluation results with individual task success rate can be found in Appendix A. We can observe that our methods already outperforms all baseline methods even without pretraining, particularly on Task 5 (*Rearrange the Restore*) and Task 17 (*Pick up then Restore*), which demonstrates the effectiveness of our multimodal prompt encoder and the

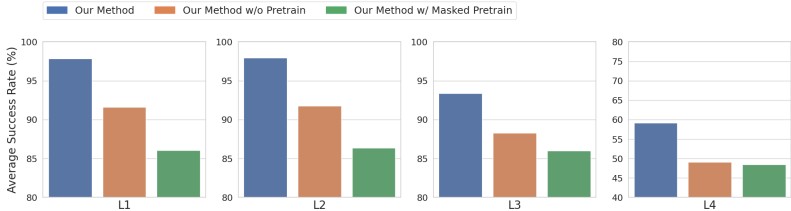

Figure 6: Ablation study on the pretraining strategy. We show that the BERT-style pretraining strategy (Our Method w/ Masked Pretrain) that performs masked action modeling does not benefit the learning of a multi-task policy to understand multimodal prompts.

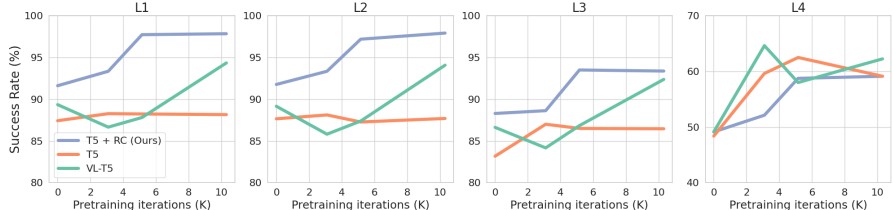

Figure 7: Ablation on the prompt encoder. We compare the performance of our methods with different prompt encoders. Our proposed T5 + RC prompt encoder that augments a pretrained T5 with a residual connection (RC) to the input visual tokens achieves a higher computational efficiency by requiring less pretraining iterations to reach a decent performance on L1, L2, and L3.

importance of modeling the dependencies between initial and target pose of the action. With pretraining, the performance of our methods improves significantly, especially on the difficult Task 9 (*Twist*) and Task 10 (*Follow Motion*). As shown in Figure 2, *Twist* requires the robot to first deduct the target rotation angles from the in-context examples before operating on the correct objects described by text. Similarly, *Follow Motion* requires the robot to deduce the actions corresponding to the image sequence in the prompt and apply them to the same object in robot's current observation. Without pretraining, models have to learn the skills for inverse dynamics prediction solely from the multi-task data, lacking enough supervision.

## 4.2 ABLATION STUDIES

We conduct extensive experiments to study how our model design and training pipeline impacts the robot manipulation, focusing on the effectiveness of our pretraining strategy and prompt encoding. We also examine the impact of data scaling and model size. Appendix A presents individual task success rate for all methods and further ablate the decoder-only architecture of our model.

**Pretraining Strategy** Figure 6 compared our pretraining strategy with a BERT-style masking prediction method (Devlin et al., 2018), which still performs the task of inverse dynamics prediction. Specially, we modify the decoding mask of the transformer to allow its attention to all future observation but mask all prompt and future action tokens. However, this pretraining strategy does not benefit the downstream multitask learning, as it does not explicitly train the model to reason the image sequences presented in the prompt.

**Multimodal Prompt Encoding** Recall that our multimodal prompt encoder (T5 + RC) augments a pretrained LM (T5-Base (Raffel et al., 2020)) with a RC to the input visual tokens. To investigate its efficacy, we compare its performance with two variants that respectively adopt a pretrained T5 and VL-T5 (Cho et al., 2021) to encode the multimodal prompt. Note that T5 is pretrained on pure text data while VL-T5 is pretrained on both vision and language data. As shown in Figure 7, our method achieves an overall better performance and computational efficiency by requiring less pretraining iterations. The comparison between the performance of T5 and VL-T5 shows that a pretrained encoder that better understands input visual tokens can benefit more from our pretraining phase.

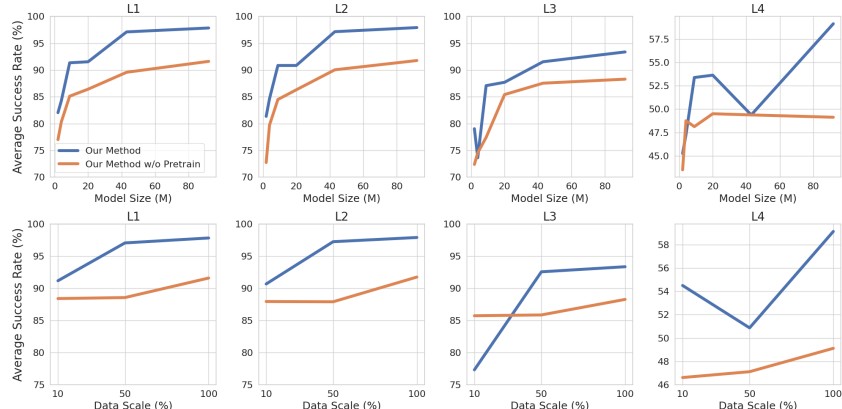

Figure 8: Ablation on model and data sizes. *Top*: For model sizes ranging from 2M to 92M, our pretraining can always learn a representation that leads to better multitask performance. *Bottom*: Model size is fixed to 92M. The benefit of pretraining increases as we increase training data size.

**Model & Data Scalability** Figure 8 illustrates the performance of our methods in relation to variations in model and data size. As the model size is scaled from 2M to 92M, we maintain a constant prompt encoder and exclude it from the parameter count, adhering to VIMA practices. Conversely, while adjusting the data size, the model remains fixed at 92M, and performance is evaluated using 10%, 50%, and 100% of the data available in VIMA-BENCH. Notably, the enhancements derived from our pretraining remain evident in low parameter regime. Additionally, a larger dataset correspondingly amplifies the performance gains achieved through our pretraining techniques.

## 4.3 EVALUATING THE IN-CONTEXT LEARNING ABILITY

Previous experiments already demonstrate the superior generalizability of our methods on L1, L2 and L3, which differs from the training tasks in object placement, combination and types. When exposed to novel tasks from L4, we expect our framework imbues the agent with a human-like intuition to learn from in-context examples. This expectation holds even if none of the training tasks explicitly present few-shot demonstrations within their prompts. To access this capability, we modify the original VIMA-BENCH by carefully constructing a new set of L4 tasks, ensuring each of the L4 tasks contain in-context examples in the prompt. Specifically, we hold out *Twist* and *Follow Order* from the training tasks, combining them with *Follow Motion* to form the new L4 task suites. The first row of Figure 2 showcases samples from these designated tasks.

As L4 tasks contain novel scenes/objects that does not exist in the training data, we leverage data augmentation during pretraining phase to improve model generalizability. Additionally, we propose *Modified FT* that randomly replace the object image in the prompt with text description provided by the VIMA-BENCH during multi-task finetuning. At inference time, we edit the prompt of *Twist* and *Follow Order* to make them closer to the pretraining prompt without adding extra task information. Appendix C provides detailed experiment setup.

As shown in Table 2, our method considerably outperforms baseline methods for the *Twist* and *Follow Motion* without decreasing its performance on L1, L2 and L3 (shown in Appendix C). *Twist* requires the model to first infer the rotation angle from prompt image sequence and then identify the target object described with text. While the imitation-learned policy (Our Method w/o Pretrain) shows limited performance on these tasks, our pretrained policy (Our Method w/ Pretrain Only) exhibits some capability, particularly in *Follow Order* which does not necessitate understanding object descriptions. However, it has difficulties with *Twist* and *Follow Motion* because it has never trained to tackle the visual and textual object. In contrast, the multi-task FT phase helps the model to understand diverse multimodal prompts and solicit its ability to translate action sequences derived from in-context examples to target objects. This is akin to the improvement seen in pretrained language models' instruction following abilities due to instruction-fining, as highlighted by (Sanh et al., 2021;

Table 2: Evaluating the in-context learning capability of the learned model. We hold out *Twist* and *Follow Order* from the training data.

| Task | T9: Twist | T10: Follow Motion | T11: Follow Order | Overall |
|------|-----------|--------------------|--------------------|---------|
| Our Method | **26.5**% | **74.0**% | 8.0 % | **36.2**% |
| Our Method w/o Modified FT | 10.0% | 43.5 % | **16.5**% | 23.3% |
| Our Method w/ Pretrain Only | 8.0 % | 2.0% | 15.5 % | 8.5 % |
| Our Method w/o Pretrain | 1.5 % | 0.5 % | 0.0% | 0.7 % |

Ouyang et al., 2022; Wei et al., 2021). Moreover, our Modified FT significantly improves model's grounding capability, contributing to a remarkable performance increase in *Twist* and *Follow Motion*.

Appendix D provides additional results, where we design 4 new tasks with in-context examples in the prompt to solidify our findings.

## 5 RELATED WORK

**Multi-Task Pretraining via Sequence Modeling.** The development of the Transformer architecture (Vaswani et al., 2017) paved the way for large-scale pretraining, which has become a standard practice to enable better generalization across different domains (Brown et al., 2020; Chen et al., 2021; Radford et al., 2021; Devlin et al., 2018; Lu et al., 2022). Specifically, these models employ the *sequential modeling* (Sutskever et al., 2014) techniques to capture temporal dependencies in the data. By training on massive web-scale data, the trained models demonstrate emergent behaviors (Brown et al., 2020; Chowdhery et al., 2022; Touvron et al., 2023), e.g., the ability to perform in-context learning. While multi-task pretraining has been extensively employed in natural language processing (NLP) and computer vision (CV), its applications in robotic systems are also gaining increasing attention (Driess et al., 2023; Brohan et al., 2022; 2023; Radosavovic et al., 2023). In our work, we pretrain our model by converting diverse robot trajectories into inverse dynamics prediction tasks, facilitating the in-context learning and multi-task performance of the our learned model.

**Multimodal Learning.** The field of multimodal learning, which focuses on integrating data from various modalities, has seen remarkable advancements (Radford et al., 2021; Wang et al., 2022; Jaegle et al., 2021). Flamingo, for instance, trains a model to generate textual completion based on multimodal prompts (Alayrac et al., 2022). The Perceiver framework (Jaegle et al., 2021) offers an adaptable method to process structured input and output. Moreover, Gato (Reed et al., 2022) introduces a versatile agent proficient in NLP, CV, and robotics. Our research tackles robot manipulation given interleaved image and text task prompt. Concurrently, MUTEX (Shah et al., 2023) learns a policy to tackle task prompts from multiple modalities (image, video, text, and speech).

**Inverse Dynamics Modeling (IDM) for Representation Learning.** IDM has proved to be an effective approach for learning from high-dimensional demonstration data. Training the model on an IDM task of predicting the agent's actions given the high-dimensional observations allows effective learning of a feature space that represents only the information relevant to the actions (Brandfonbrener et al., 2023). Pathak et al. (2017) uses IDM to generate intrinsic reward signals with self-supervision for efficient exploration. Efroni et al. (2021); Lamb et al. (2022) use a multi-step inverse dynamics model to enable representation learning robust to exogenous information. Most recently, Baker et al. (2022); Venuto et al. (2023); Thomas et al. (2023) use IDM for data-efficient multi-task pre-training on complex sequential decision-making domains. Our method leverages IDM to facilitate robot's in-context learning capability and its understanding on the transition dynamics.

## 6 CONCLUSION

In this paper, we introduce our MIDAS framework that trains a robot to comprehend multimodal prompts. The pretraining phase trains the agent to perform inverse dynamics prediction, facilitating robot's understanding of transition dynamics. To capture fine-grained visual infomration from the prompt images, we augment a pretrained LM with a RC to the object token. We further model the dependency among different action dimensions. Empirically, we establish a new state-of-the-art on the VIMA-BENCH and also demonstrate the in-context learning capability of our learned policy.

## REPRODUCIBILITY STATEMENT

We are going through internal code review and will open source our codes once it gets approved.

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

# Appendix

## A  INDIVIDUAL TASK SUCCESS RATE OF DIFFERENT METHODS

In this section, we support all experimental results in Sec. 4 of our main paper with individual task success rates for all four levels of evaluation protocol. Specifically, the results for Table 1 and the ablation on Pretraining strategies can be found in Table 3, 4, 5 and 6. Figure 7 that ablates multimodal prompt encoding is based on the results from Table 7, 8, 9 and 10. The results in Figure 8 that ablate model and data sizes are based on the results from Table 11, 12, 13, 14, 15, 16, 17, and 18.

Additionally, we further conduct an ablation study on the transformer architecture of our policy by replacing the decoder-only architecture with encoder-decoder architecture (Our Method w/ Encoder-Decoder). Experimental results in Table 3, 4, 5 and 6 show that this variant does not perform as well as our method on the L1, L2, and L3 tasks, mainly due to its inability to tackle Task 09 (*Twist*) that requires deducting rotation angles from the prompt image sequence (Figure 2). However, it achieves a superior performance on the L4 Task 10 (*Follow Motion*). We hypothesize that it is due to the limit of model capacity. This policy learns a control policy that predicts its action dependent on the object bounding box, while lacking the capability to capture fine-grained visual information that contains the information of object rotation.

Table 3: L1 level generalization results. All methods share the same amount of parameters 92M. Integers in the first row refer to indices of tasks defined in the VIMA paper (Jiang et al., 2023)

| Method | T1 | T2 | T3 | T4 | T5 | T6 | T7 | T9 | T11 | T12 | T15 | T16 | T17 | Overall |
|---|---|---|---|---|---|---|---|---|---|---|---|---|---|---|
| VIMA | 100 | 100 | 100 | 100 | 65.0 | 99.5 | 100 | 13.5 | 96.0 | 94.5 | 100 | 88.0 | 77.0 | 87.2 |
| Gato OBJ | 100 | 100 | 100 | 100 | 75.5 | 100 | 100 | 16.5 | 88.5 | 93.0 | 100 | 92.5 | 80.0 | 88.2 |
| Our Method | | | | | | | | | | | | | | |
| w/o Pretrain | 100 | 100 | 100 | 98.5 | 88.0 | 100 | 100 | 20.5 | 100 | 94.0 | 99.0 | 93.0 | 98.0 | 91.6 |
| w/ Pretrain | 98.5 | 100 | 100 | 99.5 | 94.0 | 100 | 100 | 100 | 100 | 94.0 | 95.5 | 94.0 | 96.5 | 97.8 |
| w/ Masked Pretrain | 100 | 99.5 | 100 | 99.5 | 97.5 | 99.5 | 100 | 17.5 | 74.5 | 94.5 | 97.0 | 42.5 | 96.5 | 86.0 |
| w/ Encoder-Decoder | 100 | 100 | 99.5 | 99.5 | 96.5 | 100 | 100 | 19.5 | 99.5 | 93.5 | 99.0 | 93.0 | 82.5 | 91.0 |

Table 4: L2 level generalization results. All methods share the same amount of parameters 92M. Integers in the first row refer to indices of tasks defined in the VIMA paper (Jiang et al., 2023).

| Method | T1 | T2 | T3 | T4 | T5 | T6 | T7 | T9 | T11 | T12 | T15 | T16 | T17 | Overall |
|---|---|---|---|---|---|---|---|---|---|---|---|---|---|---|
| VIMA | 100 | 100 | 100 | 99.5 | 61.0 | 100 | 100 | 12.5 | 97.5 | 95.0 | 100 | 87.5 | 77.5 | 87.0 |
| Gato OBJ | 100 | 100 | 100 | 100 | 73.0 | 100 | 100 | 17.5 | 88.5 | 95.0 | 99.0 | 94.0 | 80.5 | 88.3 |
| Our Method | | | | | | | | | | | | | | |
| w/o Pretrain | 100 | 100 | 100 | 99.0 | 87.0 | 100 | 100 | 23.5 | 100 | 94.0 | 99.5 | 92.0 | 98.0 | 91.8 |
| w/ Pretrain | 98.5 | 100 | 100 | 99.0 | 96.5 | 99.5 | 100 | 100 | 100 | 95.5 | 95.0 | 93.0 | 96.0 | 97.9 |
| w/ Masked Pretrain | 99.5 | 100 | 100 | 99.5 | 96.5 | 100 | 99.5 | 19.5 | 75.5 | 95.5 | 97.0 | 43.5 | 96.5 | 86.3 |
| w/ Encoder-Decoder | 99.5 | 100 | 99.0 | 99.5 | 96.5 | 100 | 100 | 15.5 | 99.5 | 94.0 | 98.5 | 92.0 | 82.5 | 90.5 |

Table 5: L3 level generalization results. All methods share the same amount of parameters 92M. Integers in the first row refer to indices of tasks defined in the VIMA paper (Jiang et al., 2023).

| Method | T1 | T2 | T3 | T4 | T5 | T6 | T7 | T9 | T11 | T15 | T16 | T17 | Overall |
|---|---|---|---|---|---|---|---|---|---|---|---|---|---|
| VIMA | 99.5 | 100 | 100 | 99.5 | 63.0 | 99.5 | 100 | 12.0 | 98.5 | 99.5 | 58.5 | 78.0 | 84.0 |
| Gato OBJ | 99.5 | 100 | 100 | 100 | 72.5 | 97.5 | 100 | 7.5 | 95.0 | 99.5 | 44.5 | 72.0 | 82.3 |
| Our Method | | | | | | | | | | | | | |
| w/o Pretrain | 99.5 | 100 | 100 | 100 | 90.0 | 100 | 100 | 20.5 | 100 | 99.5 | 50.5 | 99.5 | 88.3 |
| w/ Pretrain | 98.0 | 99.0 | 100 | 99.5 | 94.0 | 97.5 | 99.0 | 97.0 | 96.5 | 95.0 | 47.0 | 98.0 | 93.4 |
| w/ Masked Pretrain | 99.0 | 100 | 100 | 100 | 96.5 | 99.5 | 99.0 | 20.5 | 76.5 | 99.0 | 42.0 | 100 | 86.0 |
| w/ Encoder-Decoder | 99.0 | 99.5 | 100 | 98.5 | 95.5 | 99.5 | 98.0 | 20.0 | 100 | 95.0 | 56.0 | 86.0 | 87.2 |

Table 6: L4 level generalization results. All methods share the same amount of parameters 92M. Integers in the first row refer to indices of tasks defined in the VIMA paper (Jiang et al., 2023).

| Method | T8 | T10 | T13 | T14 | Overall |
|---|---|---|---|---|---|
| VIMA | 98.5 | 0.0 | 0.0 | 100 | 49.6 |
| Gato OBJ | 99.5 | 0.0 | 0.0 | 98.0 | 49.4 |
| Our Method | | | | | |
| w/o Pretrain | 97.0 | 0.0 | 0.0 | 99.5 | 49.1 |
| w/ Pretrain | 97.5 | 41.0 | 1.0 | 97.0 | 59.1 |
| w/ Masked Pretrain | 95.0 | 0.0 | 0.0 | 99.0 | 48.5 |
| w/ Encoder-Decoder | 96.5 | 85.5 | 0.0 | 97.0 | 69.8 |

Table 7: Comparison of the performance of our method with different multimodal prompt encoder on L1 level generalization. All methods share the same amount of parameters 92M. Integers in the first row refer to indices of tasks defined in the VIMA paper (Jiang et al., 2023)

| Pretrain iter. | Method | T1 | T2 | T3 | T4 | T5 | T6 | T7 | T9 | T11 | T12 | T15 | T16 | T17 | Overall |
|---|---|---|---|---|---|---|---|---|---|---|---|---|---|---|---|
| 0.0K | T5 + RC (Ours) | 100 | 100 | 100 | 98.5 | 88.0 | 100 | 100 | 20.5 | 100 | 94.0 | 99.0 | 93.0 | 98.0 | 91.6 |
| 0.0K | T5 | 100 | 99.5 | 98.5 | 99.0 | 84.5 | 100 | 100 | 16.0 | 100 | 94.5 | 98.0 | 49.5 | 97.0 | 87.4 |
| 0.0K | VL-T5 | 100 | 100 | 98.5 | 99.5 | 66.0 | 100 | 100 | 18.0 | 100 | 93.0 | 96.5 | 94.0 | 96.0 | 89.3 |
| 3.1K | T5 + RC (Ours) | 100 | 100 | 100 | 98.5 | 85.5 | 100 | 99.0 | 52.0 | 100 | 93.5 | 96.0 | 93.0 | 96.0 | 93.3 |
| 3.1K | T5 | 100 | 100 | 100 | 100 | 98.5 | 100 | 100 | 18.5 | 100 | 93.5 | 97.0 | 43.5 | 96.5 | 88.3 |
| 3.1K | VL-T5 | 100 | 99.5 | 100 | 100 | 70.0 | 99.5 | 100 | 23.5 | 100 | 94.5 | 99.5 | 43.5 | 96.5 | 86.7 |
| 5.2K | T5 + RC (Ours) | 100 | 100 | 100 | 99.0 | 92.0 | 99.5 | 100 | 99.5 | 100 | 95.5 | 95.0 | 93.0 | 97.0 | 97.7 |
| 5.2K | T5 | 100 | 100 | 100 | 98.5 | 98.0 | 100 | 100 | 19.5 | 100 | 94.0 | 98.5 | 42.0 | 96.5 | 88.2 |
| 5.2K | VL-T5 | 100 | 99.5 | 100 | 99.0 | 94.0 | 99.5 | 99.5 | 21.0 | 100 | 94.0 | 95.5 | 43.0 | 96.5 | 87.8 |
| 10.3K | T5 + RC (Ours) | 98.5 | 100 | 100 | 99.5 | 94.0 | 100 | 100 | 100 | 100 | 94.0 | 95.5 | 94.0 | 96.5 | 97.8 |
| 10.3K | T5 | 99.5 | 99.5 | 100 | 97.0 | 98.0 | 99.0 | 99.5 | 22.0 | 100 | 94.0 | 99.5 | 41.0 | 97.0 | 88.2 |
| 10.3K | VL-T5 | 99.5 | 99.0 | 100 | 100 | 97.5 | 99.0 | 100 | 100 | 100 | 93.5 | 98.0 | 43.0 | 97.0 | 94.3 |

Table 8: Comparison of the performance of our method with different multimodal prompt encoder on L2 level generalization. All methods share the same amount of parameters 92M. Integers in the first row refer to indices of tasks defined in the VIMA paper (Jiang et al., 2023)

| Pretrain iter. | Method | T1 | T2 | T3 | T4 | T5 | T6 | T7 | T9 | T11 | T12 | T15 | T16 | T17 | Overall |
|---|---|---|---|---|---|---|---|---|---|---|---|---|---|---|---|
| 0.0K | T5 + RC (Ours) | 100 | 100 | 100 | 99.0 | 87.0 | 100 | 100 | 23.5 | 100 | 94.0 | 99.5 | 92.0 | 98.0 | 91.8 |
| 0.0K | T5 | 99.5 | 100 | 99.0 | 99.5 | 87.0 | 100 | 99.5 | 20.0 | 100 | 93.5 | 98.5 | 48.0 | 95.0 | 87.7 |
| 0.0K | VL-T5 | 100 | 100 | 98.5 | 99.0 | 66.5 | 99.0 | 99.5 | 19.0 | 100 | 94.0 | 96.5 | 92.0 | 95.0 | 89.2 |
| 3.1K | T5 + RC (Ours) | 99.5 | 100 | 100 | 98.5 | 89.5 | 100 | 99.5 | 52.0 | 100 | 94.0 | 92.5 | 92.0 | 96.0 | 93.3 |
| 3.1K | T5 | 99.0 | 100 | 100 | 100 | 97.0 | 99.0 | 99.5 | 22.5 | 100 | 94.5 | 96.0 | 41.5 | 96.5 | 88.1 |
| 3.1K | VL-T5 | 98.0 | 99.5 | 99.5 | 98.5 | 67.5 | 99.0 | 99.5 | 24.0 | 100 | 94.0 | 96.5 | 44.0 | 95.5 | 85.8 |
| 5.2K | T5 + RC (Ours) | 100 | 100 | 100 | 97.5 | 91.0 | 98.5 | 99.5 | 99.5 | 100 | 95.5 | 93.0 | 92.5 | 96.5 | 97.2 |
| 5.2K | T5 | 98.5 | 100 | 100 | 98.0 | 96.5 | 99.0 | 98.5 | 21.5 | 100 | 94.0 | 93.5 | 40.0 | 95.0 | 87.3 |
| 5.2K | VL-T5 | 99.5 | 100 | 100 | 98.5 | 94.0 | 98.5 | 97.5 | 20.0 | 100 | 94.0 | 94.0 | 43.5 | 96.5 | 87.4 |
| 10.3K | T5 + RC (Ours) | 98.5 | 100 | 100 | 99.0 | 96.5 | 99.5 | 100 | 100 | 100 | 95.5 | 95.0 | 93.0 | 96.0 | 97.9 |
| 10.3K | T5 | 100 | 100 | 100 | 97.0 | 97.0 | 100 | 97.0 | 19.5 | 100 | 94.5 | 97.0 | 43.0 | 95.0 | 87.7 |
| 10.3K | VL-T5 | 99.0 | 99.5 | 100 | 99.5 | 97.0 | 99.0 | 99.0 | 99.5 | 100 | 94.5 | 97.5 | 43.0 | 95.5 | 94.1 |

Table 9: Comparison of the performance of our method with different multimodal prompt encoder on L3 level generalization. All methods share the same amount of parameters 92M. Integers in the first row refer to indices of tasks defined in the VIMA paper (Jiang et al., 2023)

| Pretrain iter. | Method | T1 | T2 | T3 | T4 | T5 | T6 | T7 | T9 | T11 | T15 | T16 | T17 | Overall |
|---|---|---|---|---|---|---|---|---|---|---|---|---|---|---|
| 0.0K | T5 + RC (Ours) | 99.5 | 100 | 100 | 100 | 90.0 | 100 | 100 | 20.5 | 100 | 99.5 | 50.5 | 99.5 | 88.3 |
| 0.0K | T5 | 98.5 | 98.5 | 100 | 100 | 85.5 | 99.5 | 98.5 | 19.5 | 100 | 98.5 | 42.0 | 57.5 | 83.2 |
| 0.0K | VL-T5 | 98.5 | 98.5 | 100 | 100 | 68.5 | 99.5 | 100 | 21.5 | 100 | 99.0 | 54.5 | 99.5 | 86.6 |
| 3.1K | T5 + RC (Ours) | 97.0 | 98.0 | 99.0 | 99.0 | 90.5 | 96.0 | 99.5 | 45.5 | 98.0 | 97.0 | 47.5 | 96.5 | 88.6 |
| 3.1K | T5 | 96.0 | 99.0 | 99.5 | 100 | 98.0 | 97.0 | 96.0 | 21.5 | 100 | 95.5 | 42.0 | 99.5 | 87.0 |
| 3.1K | VL-T5 | 96.5 | 97.0 | 99.5 | 99.5 | 69.0 | 94.5 | 95.0 | 21.0 | 99.5 | 96.5 | 42.0 | 100 | 84.2 |
| 5.2K | T5 + RC (Ours) | 99.5 | 99.0 | 100 | 99.0 | 93.0 | 98.0 | 99.0 | 98.0 | 98.0 | 95.5 | 46.0 | 97.0 | 93.5 |
| 5.2K | T5 | 96.5 | 96.0 | 99.5 | 100 | 97.5 | 98.5 | 97.0 | 17.5 | 100 | 97.0 | 38.5 | 100 | 86.5 |
| 5.2K | VL-T5 | 96.0 | 98.5 | 99.5 | 100 | 96.5 | 95.5 | 95.5 | 21.0 | 100 | 98.0 | 41.5 | 100 | 86.8 |
| 10.3K | T5 + RC (Ours) | 98.0 | 99.0 | 100 | 99.5 | 94.0 | 97.5 | 99.0 | 97.0 | 96.5 | 95.0 | 47.0 | 98.0 | 93.4 |
| 10.3K | T5 | 99.0 | 97.5 | 100 | 99.5 | 96.5 | 96.0 | 95.0 | 15.5 | 100 | 95.5 | 43.5 | 99.5 | 86.5 |
| 10.3K | VL-T5 | 98.0 | 97.0 | 100 | 99.5 | 96.0 | 97.0 | 96.5 | 84.5 | 100 | 99.5 | 41.0 | 99.5 | 92.4 |

Table 10: Comparison of the performance of our method with different multimodal prompt encoder on L4 level generalization. All methods share the same amount of parameters 92M. Integers in the first row refer to indices of tasks defined in the VIMA paper (Jiang et al., 2023)

| Pretrain iter. | Method | T8 | T10 | T13 | T14 | Overall |
|---|---|---|---|---|---|---|
| 0.0K | T5 + RC (Ours) | 97.0 | 0.0 | 0.0 | 99.5 | 49.1 |
| 0.0K | T5 | 95.0 | 0.0 | 0.0 | 98.5 | 48.4 |
| 0.0K | VL-T5 | 99.0 | 0.0 | 0.0 | 97.5 | 49.1 |
| 3.1K | T5 + RC (Ours) | 97.5 | 12.5 | 0.0 | 98.5 | 52.1 |
| 3.1K | T5 | 98.0 | 45.0 | 0.0 | 95.5 | 59.6 |
| 3.1K | VL-T5 | 98.5 | 64.0 | 0.0 | 96.0 | 64.6 |
| 5.2K | T5 + RC (Ours) | 98.0 | 40.5 | 0.0 | 96.5 | 58.8 |
| 5.2K | T5 | 98.5 | 55.5 | 0.0 | 96.0 | 62.5 |
| 5.2K | VL-T5 | 98.0 | 37.5 | 0.0 | 96.5 | 58.0 |
| 10.3K | T5 + RC (Ours) | 97.5 | 41.0 | 1.0 | 97.0 | 59.1 |
| 10.3K | T5 | 98.5 | 39.5 | 0.0 | 98.5 | 59.1 |
| 10.3K | VL-T5 | 97.5 | 53.5 | 0.0 | 98.0 | 62.3 |

Table 11: Comparison of the performance of our method with different model sizes ranging from 2M to 92M on L1 level generalization results. Integers in the first row refer to indices of tasks defined in the VIMA paper (Jiang et al., 2023)

| Model size. | Method | T1 | T2 | T3 | T4 | T5 | T6 | T7 | T9 | T11 | T12 | T15 | T16 | T17 | Overall |
|---|---|---|---|---|---|---|---|---|---|---|---|---|---|---|---|
| 2M | Ours w/o Pretrain | 100 | 98.5 | 99.0 | 89.5 | 48.5 | 100 | 100 | 19.5 | 97.0 | 91.0 | 98.0 | 36.0 | 24.0 | 77.0 |
| 2M | Ours | 99.5 | 99.0 | 97.5 | 99.0 | 67.5 | 100 | 99.5 | 18.5 | 91.5 | 93.0 | 99.0 | 38.0 | 64.5 | 82.0 |
| 4M | Ours w/o Pretrain | 100 | 100 | 99.5 | 97.0 | 55.0 | 100 | 100 | 18.0 | 96.0 | 95.0 | 99.5 | 44.0 | 40.0 | 80.3 |
| 4M | Ours | 100 | 100 | 86.5 | 99.0 | 63.5 | 99.5 | 100 | 20.5 | 92.0 | 95.5 | 98.0 | 83.5 | 57.0 | 84.2 |
| 9M | Ours w/o Pretrain | 100 | 100 | 96.0 | 99.0 | 57.0 | 100 | 100 | 23.0 | 98.0 | 94.0 | 98.5 | 47.0 | 94.0 | 85.1 |
| 9M | Ours | 100 | 100 | 99.0 | 99.0 | 87.0 | 100 | 100 | 19.0 | 100 | 95.5 | 98.5 | 92.5 | 97.0 | 91.3 |
| 20M | Ours w/o Pretrain | 100 | 100 | 100 | 98.5 | 67.5 | 100 | 100 | 30.5 | 98.5 | 95.0 | 99.0 | 49.5 | 85.0 | 86.4 |
| 20M | Ours | 100 | 100 | 100 | 97.0 | 90.0 | 100 | 99.5 | 19.0 | 100 | 94.0 | 99.5 | 93.5 | 97.5 | 91.5 |
| 43M | Ours w/o Pretrain | 100 | 100 | 100 | 98.5 | 67.0 | 100 | 100 | 17.0 | 100 | 94.0 | 99.0 | 92.5 | 96.5 | 89.6 |
| 43M | Ours | 99.5 | 100 | 99.5 | 95.5 | 89.0 | 97.5 | 100 | 100 | 100 | 94.5 | 96.0 | 94.5 | 96.5 | 97.1 |
| 92M | Ours w/o Pretrain | 100 | 100 | 100 | 98.5 | 88.0 | 100 | 100 | 20.5 | 100 | 94.0 | 99.0 | 93.0 | 98.0 | 91.6 |
| 92M | Ours | 98.5 | 100 | 100 | 99.5 | 94.0 | 100 | 100 | 100 | 100 | 94.0 | 95.5 | 94.0 | 96.5 | 97.8 |

Table 12: Comparison of the performance of our method with different model sizes ranging from 2M to 92M on L2 level generalization results. Integers in the first row refer to indices of tasks defined in the VIMA paper (Jiang et al., 2023)

| Model size. | Method | T1 | T2 | T3 | T4 | T5 | T6 | T7 | T9 | T11 | T12 | T15 | T16 | T17 | Overall |
|---|---|---|---|---|---|---|---|---|---|---|---|---|---|---|---|
| 2M | Ours w/o Pretrain | 95.5 | 84.5 | 99.0 | 87.0 | 50.0 | 96.5 | 91.0 | 21.0 | 97.0 | 91.0 | 88.0 | 33.5 | 11.5 | 72.7 |
| 2M | Ours | 99.5 | 98.5 | 98.5 | 98.5 | 59.0 | 100 | 98.5 | 20.5 | 92.0 | 92.5 | 99.0 | 39.5 | 61.5 | 81.3 |
| 4M | Ours w/o Pretrain | 99.0 | 98.5 | 100 | 97.0 | 55.0 | 99.5 | 98.5 | 21.0 | 96.5 | 95.5 | 97.0 | 44.0 | 35.0 | 79.7 |
| 4M | Ours | 100 | 100 | 87.0 | 99.0 | 67.5 | 99.5 | 99.5 | 19.0 | 92.5 | 95.5 | 97.0 | 84.0 | 60.0 | 84.7 |
| 9M | Ours w/o Pretrain | 100 | 100 | 96.5 | 98.5 | 58.0 | 99.5 | 99.0 | 25.5 | 97.5 | 94.5 | 94.5 | 47.0 | 88.0 | 84.5 |
| 9M | Ours | 100 | 100 | 99.5 | 98.5 | 86.5 | 99.5 | 100 | 19.0 | 100 | 94.5 | 97.0 | 91.5 | 95.0 | 90.8 |
| 20M | Ours w/o Pretrain | 100 | 100 | 100 | 98.5 | 72.0 | 100 | 100 | 29.5 | 98.0 | 95.5 | 99.0 | 46.0 | 83.5 | 86.3 |
| 20M | Ours | 100 | 100 | 100 | 97.0 | 86.5 | 99.0 | 99.0 | 19.5 | 100 | 95.0 | 97.0 | 91.5 | 96.5 | 90.8 |
| 43M | Ours w/o Pretrain | 100 | 100 | 100 | 98.5 | 72.5 | 100 | 100 | 18.5 | 100 | 93.5 | 99.5 | 92.0 | 96.0 | 90.0 |
| 43M | Ours | 99.0 | 100 | 100 | 97.0 | 90.5 | 98.0 | 100 | 99.5 | 100 | 95.5 | 94.0 | 93.0 | 96.5 | 97.2 |
| 92M | Ours w/o Pretrain | 100 | 100 | 100 | 99.0 | 87.0 | 100 | 100 | 23.5 | 100 | 94.0 | 99.5 | 92.0 | 98.0 | 91.8 |
| 92M | Ours | 98.5 | 100 | 100 | 99.0 | 96.5 | 99.5 | 100 | 100 | 100 | 95.5 | 95.0 | 93.0 | 96.0 | 97.9 |

Table 13: Comparison of the performance of our method with different model sizes ranging from 2M to 92M on L3 level generalization results. Integers in the first row refer to indices of tasks defined in the VIMA paper (Jiang et al., 2023)

| Model size. | Method | T1 | T2 | T3 | T4 | T5 | T6 | T7 | T9 | T11 | T15 | T16 | T17 | Overall |
|---|---|---|---|---|---|---|---|---|---|---|---|---|---|---|
| 2M | Ours w/o Pretrain | 97.0 | 91.0 | 100 | 92.5 | 46.0 | 96.5 | 95.5 | 15.5 | 95.5 | 95.0 | 36.0 | 8.0 | 72.4 |
| 2M | Ours | 99.0 | 98.5 | 99.5 | 99.5 | 67.0 | 99.5 | 99.5 | 16.5 | 83.5 | 98.5 | 33.5 | 54.0 | 79.0 |
| 4M | Ours w/o Pretrain | 98.5 | 98.0 | 100 | 94.5 | 53.5 | 95.0 | 99.0 | 17.0 | 99.0 | 87.5 | 47.0 | 5.5 | 74.5 |
| 4M | Ours | 95.5 | 99.0 | 75.5 | 95.5 | 61.0 | 95.5 | 96.5 | 19.0 | 92.5 | 79.5 | 42.0 | 32.0 | 73.6 |
| 9M | Ours w/o Pretrain | 93.5 | 97.5 | 96.0 | 100 | 64.0 | 95.0 | 96.0 | 17.5 | 97.5 | 85.0 | 43.0 | 44.0 | 77.4 |
| 9M | Ours | 97.0 | 97.0 | 100 | 96.5 | 86.0 | 99.0 | 99.0 | 23.5 | 98.5 | 96.0 | 52.5 | 100 | 87.1 |
| 20M | Ours w/o Pretrain | 99.5 | 100 | 100 | 100 | 71.5 | 100 | 100 | 26.0 | 98.5 | 98.5 | 43.5 | 87.5 | 85.4 |
| 20M | Ours | 97.0 | 97.0 | 99.5 | 99.5 | 89.0 | 98.0 | 98.0 | 24.0 | 100 | 99.0 | 53.5 | 98.0 | 87.7 |
| 43M | Ours w/o Pretrain | 99.5 | 100 | 100 | 100 | 74.0 | 100 | 99.5 | 25.0 | 100 | 99.5 | 54.0 | 99.0 | 87.5 |
| 43M | Ours | 95.0 | 98.0 | 99.5 | 96.5 | 86.0 | 95.5 | 96.5 | 97.0 | 99.5 | 96.0 | 40.0 | 99.0 | 91.5 |
| 92M | Ours w/o Pretrain | 99.5 | 100 | 100 | 100 | 90.0 | 100 | 100 | 20.5 | 100 | 99.5 | 50.5 | 99.5 | 88.3 |
| 92M | Ours | 98.0 | 99.0 | 100 | 99.5 | 94.0 | 97.5 | 99.0 | 97.0 | 96.5 | 95.0 | 47.0 | 98.0 | 93.4 |

Table 14: Comparison of the performance of our method with different model sizes ranging from 2M to 92M on L4 level generalization results. Integers in the first row refer to indices of tasks defined in the VIMA paper (Jiang et al., 2023)

| Model size. | Method | T8 | T10 | T13 | T14 | Overall |
|---|---|---|---|---|---|
| 2M | Ours w/o Pretrain | 78.5 | 0.0 | 0.0 | 95.5 | 43.5 |
| 2M | Ours | 47.5 | 35.5 | 0.5 | 97.5 | 45.2 |
| 4M | Ours w/o Pretrain | 99.5 | 0.0 | 0.0 | 95.5 | 48.8 |
| 4M | Ours | 96.0 | 0.5 | 0.0 | 92.5 | 47.2 |
| 9M | Ours w/o Pretrain | 96.5 | 1.0 | 0.0 | 95.0 | 48.1 |
| 9M | Ours | 99.5 | 15.5 | 0.5 | 98.0 | 53.4 |
| 20M | Ours w/o Pretrain | 99.0 | 0.0 | 0.0 | 99.0 | 49.5 |
| 20M | Ours | 97.0 | 22.0 | 0.0 | 95.5 | 53.6 |
| 43M | Ours w/o Pretrain | 99.0 | 0.0 | 0.0 | 98.5 | 49.4 |
| 43M | Ours | 95.5 | 6.0 | 0.0 | 96.0 | 49.4 |
| 92M | Ours w/o Pretrain | 97.0 | 0.0 | 0.0 | 99.5 | 49.1 |
| 92M | Ours | 97.5 | 41.0 | 1.0 | 97.0 | 59.1 |

Table 15: Comparison of the performance of our method with different scales of training data on L1 level generalization results. Integers in the first row refer to indices of tasks defined in the VIMA paper (Jiang et al., 2023)

| Data Size | Method | T1 | T2 | T3 | T4 | T5 | T6 | T7 | T9 | T11 | T12 | T15 | T16 | T17 | Overall |
|---|---|---|---|---|---|---|---|---|---|---|---|---|---|---|---|
| 10% | Ours w/o Pretrain | 100 | 98.5 | 96.5 | 97.0 | 74.0 | 97.5 | 100 | 19.0 | 100 | 93.0 | 93.0 | 88.0 | 93.0 | 88.4 |
| 10% | Ours | 100 | 99.5 | 96.5 | 89.0 | 65.5 | 98.0 | 98.5 | 73.5 | 97.5 | 93.5 | 92.0 | 89.0 | 93.0 | 91.2 |
| 50% | Ours w/o Pretrain | 100 | 99.5 | 97.5 | 98.0 | 74.0 | 99.5 | 99.5 | 20.0 | 100 | 92.5 | 98.0 | 82.0 | 91.0 | 88.6 |
| 50% | Ours | 100 | 99.5 | 98.5 | 98.0 | 86.5 | 99.5 | 99.5 | 98.5 | 100 | 93.5 | 96.5 | 95.5 | 96.5 | 97.1 |
| 100% | Ours w/o Pretrain | 100 | 100 | 100 | 98.5 | 88.0 | 100 | 100 | 20.5 | 100 | 94.0 | 99.0 | 93.0 | 98.0 | 91.6 |
| 100% | Ours | 98.5 | 100 | 100 | 99.5 | 94.0 | 100 | 100 | 100 | 100 | 94.0 | 95.5 | 94.0 | 96.5 | 97.8 |

Table 16: Comparison of the performance of our method with different scales of training data on L2 level generalization results. Integers in the first row refer to indices of tasks defined in the VIMA paper (Jiang et al., 2023)

| Data Size | Method | T1 | T2 | T3 | T4 | T5 | T6 | T7 | T9 | T11 | T12 | T15 | T16 | T17 | Overall |
|---|---|---|---|---|---|---|---|---|---|---|---|---|---|---|---|
| 10% | Ours w/o Pretrain | 99.5 | 99.0 | 97.0 | 95.5 | 72.5 | 97.5 | 99.5 | 20.5 | 98.5 | 93.0 | 91.5 | 88.5 | 91.0 | 88.0 |
| 10% | Ours | 99.0 | 99.5 | 94.5 | 90.0 | 62.5 | 98.5 | 99.0 | 77.5 | 98.5 | 94.0 | 90.0 | 87.0 | 89.0 | 90.7 |
| 50% | Ours w/o Pretrain | 98.5 | 100 | 97.0 | 98.0 | 72.0 | 99.5 | 99.5 | 16.5 | 100 | 91.5 | 97.5 | 84.5 | 88.5 | 87.9 |
| 50% | Ours | 100 | 100 | 99.0 | 97.5 | 88.5 | 99.0 | 99.0 | 98.5 | 100 | 95.0 | 96.0 | 95.5 | 96.5 | 97.3 |
| 100% | Ours w/o Pretrain | 100 | 100 | 100 | 99.0 | 87.0 | 100 | 100 | 23.5 | 100 | 94.0 | 99.5 | 92.0 | 98.0 | 91.8 |
| 100% | Ours | 98.5 | 100 | 100 | 99.0 | 96.5 | 99.5 | 100 | 100 | 100 | 95.5 | 95.0 | 93.0 | 96.0 | 97.9 |

Table 17: Comparison of the performance of our method with different scales of training data on L3 level generalization results. Integers in the first row refer to indices of tasks defined in the VIMA paper (Jiang et al., 2023)

| Data Size | Method | T1 | T2 | T3 | T4 | T5 | T6 | T7 | T9 | T11 | T15 | T16 | T17 | Overall |
|---|---|---|---|---|---|---|---|---|---|---|---|---|---|---|
| 10% | Ours w/o Pretrain | 98.0 | 97.0 | 97.5 | 98.5 | 74.5 | 97.5 | 99.0 | 18.0 | 100 | 96.5 | 53.5 | 99.0 | 85.8 |
| 10% | Ours | 90.0 | 93.5 | 98.5 | 93.0 | 71.0 | 90.0 | 90.5 | 56.0 | 90.0 | 83.0 | 52.0 | 20.5 | 77.3 |
| 50% | Ours w/o Pretrain | 97.5 | 99.5 | 99.0 | 99.5 | 70.5 | 98.5 | 99.0 | 19.0 | 100 | 97.0 | 57.5 | 93.5 | 85.9 |
| 50% | Ours | 97.0 | 97.5 | 99.0 | 99.5 | 86.0 | 97.5 | 96.5 | 95.5 | 98.0 | 97.0 | 47.5 | 100 | 92.6 |
| 100% | Ours w/o Pretrain | 99.5 | 100 | 100 | 100 | 90.0 | 100 | 100 | 20.5 | 100 | 99.5 | 50.5 | 99.5 | 88.3 |
| 100% | Ours | 98.0 | 99.0 | 100 | 99.5 | 94.0 | 97.5 | 99.0 | 97.0 | 96.5 | 95.0 | 47.0 | 98.0 | 93.4 |

Table 18: Comparison of the performance of our method with different scales of training data on L4 level generalization results. Integers in the first row refer to indices of tasks defined in the VIMA paper (Jiang et al., 2023)

| Data Size | Method | T8 | T10 | T13 | T14 | Overall |
|---|---|---|---|---|---|
| 10% | Ours w/o Pretrain | 92.0 | 0.0 | 0.0 | 94.5 | 46.6 |
| 10% | Ours | 91.0 | 39.0 | 0.0 | 88.0 | 54.5 |
| 50% | Ours w/o Pretrain | 91.5 | 0.0 | 0.0 | 97.0 | 47.1 |
| 50% | Ours | 95.0 | 12.5 | 0.0 | 96.0 | 50.9 |
| 100% | Ours w/o Pretrain | 97.0 | 0.0 | 0.0 | 99.5 | 49.1 |
| 100% | Ours | 97.5 | 41.0 | 1.0 | 97.0 | 59.1 |

## B  PSEUDO-CODES & TRAINING DETAILS

---

**Algorithm 1** Robot Control with multimodal prompts through pretraining and multitask FT

---

**Input**: Dataset $\mathcal{D} = \{\zeta_1, \zeta_2, \ldots\}$, policy parameter $\theta$, number of pretraining iterations $N_{\text{pretrain}}$, number of multi-task imitation finetuning iterations $N_{\text{FT}}$

1: **for** $i = 1, \ldots, N_{\text{pretrain}}$ **do**
2:     Sample a mini-batch $\mathcal{B}$ from $\mathcal{D}$
3:     Minimize $L_{\text{pretrain}}(\theta)$ defined in Eq. 3 on $\mathcal{B}$
4: **end for**
5: **for** $i = 1, \ldots, N_{\text{FT}}$ **do**
6:     Sample a mini-batch $\mathcal{B}$ from $\mathcal{D}$
7:     Minimize $L_{\text{Imitatation}}(\theta)$ defined in Eq. 4 on $\mathcal{B}$
8: **end for**

---

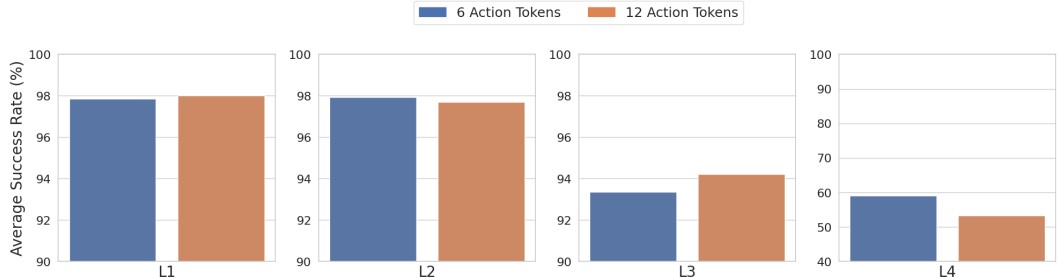

Figure 9: Ablation on the number of action tokens.

Algorithm 1 presents the pseudo-codes for the training pipeline, which includes a pretraining phase and a multi-task FT phase. We set our training HP following the recipe provided by VIMA, which open-sourced its policy architectures without providing the training codes. We conduct our experiments on cluster nodes, each with 8 NVIDIA-A10G. Table 19 presents the HP for our training pipeline. As we build our policy based on the VIMA Policy, we refer interested readers to Tables 2 and 3 in Appendix C of VIMA paper (Jiang et al., 2023) for all model parameters.

Additionally, the action space $\mathcal{A}$ includes initial pose $\mathcal{T}_{\text{initial}} \in \mathcal{R}^6$ and $\mathcal{T}_{\text{target}} \in \mathcal{R}^6$. Each pose is a 6-dimension vector with 2 for xy position and 4 for rotation represented in quaternion. Since the VIMA-BENCH focuses on tabletop manipulation, the rotation quaternion of $\mathcal{T}_{\text{initial}}$ is always a constant vector. So is the first two dimensions of the rotation quaternion of $\mathcal{T}_{\text{initial}}$. Therefore, we only tokenize the other 6 action dimensions to improve computational efficiency. Thus, each action worth 6 tokens. Moreover, we conduct an ablation study to show that this choice will not affect the task success rate. As shown in Figure 9, modeling each of the 12 action dimensions as a single token achieves almost the same performance as modeling the 6 active action dimensions.

Table 19: Hyper-parameters for our training pipeline

| Phase | Hyperparameter | Value |
|---|---|---|
| Shared | Learning Rate (LR) | 1e-4 |
| | Minimum LR | 1e-7 |
| | Warmup Steps | 7K |
| | Weight Decay | 0 |
| | Dropout | 0.1 |
| | Gradient Clip Threshold | 1.0 |
| | Optimizer | AdamW (Loshchilov & Hutter, 2017) |
| | Batch Size | 128 |
| | Iterations per epochs | 5158 |
| Pretrain | Training epochs | 20 |
| | Training iterations $N_{\text{pretrain}}$ | $20 \times$ Iterations per epochs = 103160 |
| | LR Cosine Annealing Steps | $N_{\text{pretrain}}$ - Warmup Steps = 96160 |
| Finetune | LR Cosine Annealing Steps | 17K |
| | Training epochs | 10 |
| | Training iterations $N_{\text{FT}}$ | $10 \times$ Iterations per epochs = 51580 |

## C   DETAILS OF EVALUATING THE IN-CONTEXT LEARNING ABILITY

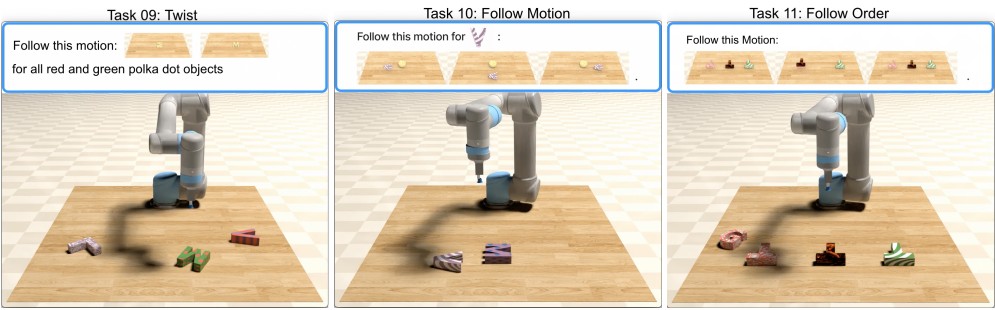

Figure 10: The new set of L4 tasks with in-context examples and modified prompts.

We provide training details for the experiments conducted in Sec. 4.3 by introducing the data augmentation strategies, pretraining, our *Modified FT*, and how we edit the task prompt for Task 09 (*Twist*) and Task 10 (*Follow Order*). Moreover, the L1, L2 and L3 success rate in this settings are given by 97.6%, 97.7%, and 93.0%, respectively.

**Data Augmentation** To improve the generalizability of the pretrained model, we randomly apply the standard random data augmentation techniques, including Color Jitter and Gray Scale (He et al., 2020) to the prompt images. Since we adopt an object-centric representation, we randomly shift the bounding box location for all objects in the whole trajectory with the same constant value. Note that we only augment the prompt images without modifying the observation images.

**Pretraining** We empirically find that further dividing the pretraining phase into two steps can improve the performance. We first pretrain a policy for 20 epochs and only extract the object encoder from it. Next, we use the pretrained object encoder to initialize another policy and pretrain it for 5 epochs. And the FT phase remains unchanged.

**Modified FT** To improve the model's ability to understand both visual and textual object descriptions, we randomly replace the object images in the multimodal prompts with text descriptions during multi-task FT. For example, the task prompt for *Follow Motion* in Figure 10 can be rephrased as

> Follow this motion for the white and purple striped V: $\{frame_1\}$, $\{frame_2\}$, $\{frame_3\}$.

Note that only object images will be converted into text descriptions. Images depicted the scene, e.g., $frame_1$, $frame_2$, $frame_3$, will never be converted to text. We randomly apply this operation to the task prompt of the pretraining tasks during the FT phase.

**Edit Prompts** As shown in Figure 10, we modify the task prompt for both *Twist* and *Follow order* to make them similar to the pretraining prompts. Specifically, the task prompt for *Twist* is modified as below

> **Original**: "Twist" is defined as rotating object a specific angle. For examples: From $\{before\_twist_1\}$ to $\{after\_twist_1\}$. From $\{before\_twist_2\}$ to $\{after\_twist_2\}$. From $\{before\_twist_3\}$ to $\{after\_twist_3\}$. Now twist all [TEXT OBJ DESCRIPTION] objects.
>
> **Modified**: Follow this motion: $\{before\_twist_1\}$ to $\{after\_twist_1\}$ for all [TEXT OBJ DESCRIPTION] objects.

Similarly, the task prompt for *Follow Order* is modified as below:

> **Original**: Stack objects in this order $\{frame_1\}$, $\{frame_2\}$, $\{frame_3\}$.
>
> **Modified**: Follow this motion: $\{frame_1\}$, $\{frame_2\}$, $\{frame_3\}$.

# D ADDITIONAL L4 UNSEEN TASKS WITH IN-CONTEXT EXAMPLES

We augment the L4 task suite of VIMA-BENCH by designing 4 new tasks with in-context examples provided in the prompt. These tasks are within the *One-shot Video Imitation* category of VIMA-BENCH (Appendix B.4, Jiang et al. (2023)). Next, we will first provide the task definitions. Then, we take our policy that is trained on the full data of VIMA-BENCH and evaluate it on these tasks. Notably, we never use trajectories collected from these tasks to train our policy.

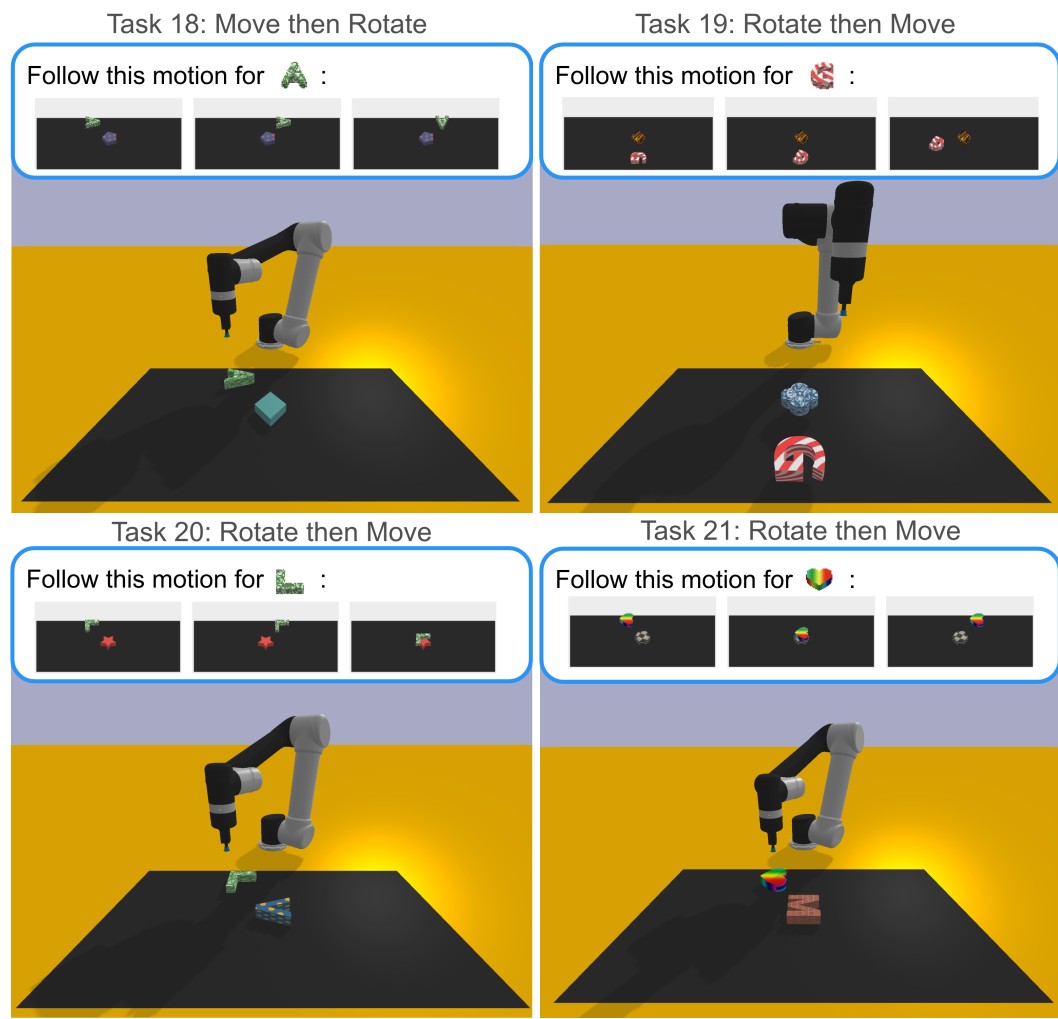

Figure 11: Task samples from our designed tasks. Each task is paired with in-context demonstration in the prompt.

## D.1 TASK DEFINITION

To evaluate the in-context learning ability of a policy, we design four tasks to incorporate a demonstration trajectory in the task prompt. Specifically, these tasks share the same prompt template

Follow this motion for {target object}: {frame$_1$}, {frame$_2$}, {frame$_3$}.

Note that we did not inject language variety to the task prompt, **as we can always paraphrase the task prompt to the unified prompt defined above given demonstration trajectory**.

Image placeholder {target object} is the target object to be manipulated and {{frame$_i$}} a set of workspace-like scene placeholders to represent a video trajectory. Distractor objects are spawned

at the center of the workspace and the prompt video. However, the distractor in the workspace is different from the distractor in the prompt video. The initial position of the target object matches that in $\{\text{frame}_1\}$.

**Task 18: Move then Rotate**. The robot should first move the target object to a specific location and then rotate the target object by a certain degree, according to the demonstration trajectory.

**Task 19: Rotate then Move**. The robot should first rotate the target object by a certain degree and then move the target object to a specific location according to the demonstration trajectory.

**Task 20: Move then Stack**. The robot should first stack the target object on the distractor and then move the target object to a specific location according to the demonstration trajectory.

**Task 21: Rotate then Move**. The robot should first move the target object to a specific location and then stack the target object on the distractor according to the demonstration trajectory.

## D.2 EXPERIMENTAL RESULTS ON THE UNSEEN TASKS

Table 20: Evaluating the in-context learning capability of the learned model on the four unseen tasks proposed in Appendix D.1. All policies are trained on the full data of VIMA-BENCH.

| Method | Task 18 | Task 19 | Task 20 | Task 21 | Overall |
|---|---|---|---|---|---|
| Our Method | **13.5**% | **14.5**% | **22.0**% | **10.0**% | **15.0**% |
| Our Method w/ Pretrain Only | 4.5% | 0.5% | 20.5% | 1.5% | 5.3% |
| Our Method w/o Pretrain | 0.0% | 0.0% | 0.0% | 0.5% | 0.1% |
| VIMA | 0.0% | 0.0% | 0.0% | 0.5% | 0.1% |

We take policies trained on the full VIMA-BENCH data and directly compare their performance on these four new tasks. As shown in Table 20, Our Method significantly outperforms the baseline methods. On the other hand, the VIMA policy struggles to perform well on these tasks, showing its inability to learn from the in-context demonstration. Moreover, comparing the performance of Our Method with Our Method w/ Pretrain Only, we can conclude that our two-stage training pipeline produces a better in-context learner.

