# OpenReview forum: "Mastering Robot Manipulation with Multimodal Prompts through Pretraining and Multi-task Fine-tuning"
_ICLR.cc/2024/Conference — Submitted to ICLR 2024_

### Official Review · Reviewer_yFyT · 2023-10-28

**Soundness:** 3 good
**Presentation:** 4 excellent
**Contribution:** 2 fair
**Rating:** 6
**Confidence:** 5

**Summary:**

This paper proposes new learning methods to master simulated tabletop robot manipulation from multi-modal prompts. Specifically, their method involves two stages, first inverse-dynamics pretraining then multi-task finetuning. State-of-the-art results are demonstrated on the multimodal prompt benchmark VIMA-BENCH. Furthermore, authors conducted ablation studies to justify the effectiveness of design choices and showcase in-context learning ability achieved by the trained model.

**Strengths:**

- The proposed method is effective, as demonstrated by its new SOTA performance on VIMA-BENCH.
- Comprehensive ablation studies draw insights into the effectiveness of proposed method.
- Demonstrated in-context learning ability described in Section 4.3 is interesting and impressive.
- The paper is well-written and presented.

**Weaknesses:**

- Albeit the method is interesting and demonstrated improvement is impressive, the proposed method is only evaluated on a single benchmark. It would be more solid if authors cloud show similar improvement on other robot learning benchmarks such as RLBench (James et al., 2020).
- It's totally legitimate for the authors to argue other benchmarks do not support multimodal prompts. In that case, I would encourage authors to extend existing VIMA-BENCH by adding more representative tasks to show the in-context learning ability of models trained with the proposed method.
- Although this paper is not designed to address real-robot manipulation, showing proof-of-concept demos would justify the feasibility of applying this method on real hardware.
- Missing citations. Authors are encouraged to discuss the following recent related work:

Radosavovic et al., Robot Learning with Sensorimotor Pre-training, arXiv 2023.

Shah et al., MUTEX: Learning Unified Policies from Multimodal Task Specifications, CoRL 2023.

## References
James et al., RLBench: The Robot Learning Benchmark & Learning Environment, IEEE Robotics and Automation Letters 2020.

**Questions:**

To encode multimodal prompts, the introduced RC provides a direct connection between input embeddings and LM's output embeddings. With this shortcut, is there any performance difference between LMs with varying depth?

---

> ### Author Response · Authors · 2023-11-16
> **Response to reviewer yFyT**
>
> We are grateful for the positive and detailed feedback from the reviewer. Please find our responses below. We hope that the following response will further clarify any concerns and assist in advocating for the acceptance of our paper.
>
> > It would be more solid if authors cloud show similar improvement on other robot learning benchmarks such as RLBench (James et al., 2020).
>
> As mentioned by the reviewer, RLBench does not support multimodal prompts. Thus, we choose to extend the existing VIMA-BENCH by adding more tasks to showcase the in-context learning ability of our models.
>
> > I would encourage authors to extend existing VIMA-BENCH by adding more representative tasks to show the in-context learning ability of models trained with the proposed method.
>
> We thank the reviewers for the constructive comments! We designed 4 new tasks with in-context learning examples in the task prompt. We evaluate our policy that is trained on the full VIMA-BENCH data on these 4 tasks and compare it with the VIMA policy. The table below shows that our method demonstrates superior performance over all baseline methods. On the other hand, the VIMA policy struggles with these tasks, revealing its inability to learn from the in-context demonstration. More experiment details, including task definitions, can be found in Appendix D.
>
> | Method                        | Move Then Rotate | Rotate Then Move | Move Then Stack | Stack Then Move | Average Success Rate |
> |-------------------------------|------------------|------------------|-----------------|-----------------|----------------------|
> | Our Method                    | **13.5%**         | **14.5%**        | **22.0%**       | **10.0%**       | **15.0%**            |
> | Our Method w/ Pretrain Only   | 4.5%             | 0.5%             | 20.5%           | 1.5%            | 5.3%                 |
> | Our Method w/o Pretrain       | 0.0%             | 0.0%             | 0.0%            | 0.5%            | 0.1%                 |
> | VIMA                          | 0.0%             | 0.0%             | 0.0%            | 0.5%            | 0.1%                 |
>
> > Missing citations. Authors are encouraged to discuss the following recent related work
> >
>
> Thank you for bringing up these two works! We have updated our manuscript to cite and discuss these works in the related work section.
>
> > Although this paper is not designed to address real-robot manipulation, showing proof-of-concept demos would justify the feasibility of applying this method on real hardware.
>
> We appreciate the reviewer's suggestion. We currently do not have access to a real robot and thus cannot conduct a demonstration. Nevertheless, we assume the same action space as in Transporter [1] and CLIPort [2] and share similar settings except for tackling multimodal prompts. These alignments suggest strong potential for real-world deployment of our method. We believe our contributions should still be valid in the absence of a real-robot demonstration.
>
> > With this shortcut, is there any performance difference between LMs with varying depth?
>
> We evaluate our methods by replacing the T5-base with T5-small. The experiment result is shown below. We can see that the performance difference is minimal.
>
> | Model      | L1    | L2    | L3    | L4    |
> |------------|-------|-------|-------|-------|
> | T5-base    | 97.8% | 97.9% | 93.4% | 59.1% |
> | T5-small   | 97.9% | 98.0% | 92.7% | 50.0% |
>
> [1] Zeng et al., Transporter Networks: Rearranging the Visual World for Robotic Manipulation, CoRL 2020.
>
> [2] Shridhar et al., CLIPORT: What and Where Pathways for Robotic Manipulation, CoRL 2021.

---

> > ### Comment · Reviewer_yFyT · 2023-11-22
> >
> > Thanks authors for conducting extra experiments to address my questions. I would like to keep my original rating.

---

> ### Author Response · Authors · 2023-11-21
> **Follow up the disccusion**
>
> Dear Reviewer yFyT,
>
> Thank you again for your time and effort. Your feedback has been valuable in helping us clarify, improve, and refine our work. We have carefully addressed your comments in our authors' responses to improve the quality of our paper. We thus kindly request that you take a moment to revisit our paper and consider the changes we have made. We hope our clarifications can help you better advocate for our paper's acceptance.
>
> Best regards,
>
> The authors

---

### Official Review · Reviewer_Y3nY · 2023-10-31

**Soundness:** 3 good
**Presentation:** 3 good
**Contribution:** 2 fair
**Rating:** 5
**Confidence:** 4

**Summary:**

This work proposes a new method to learn multi-model prompt robot policy. The  main differences from a prior work, VIMA, are the following:

1. Have a pre-training phrase that pretrains on prompts asking the robot to follow a certain motion.
2. A new encoding method to encode multi-model prompts.
3. A method to model the dependency among action dimensions.

**Strengths:**

- The pretraining method makes sense in that it uses the implicit motion data in each trajectory as the training signal.
- The new prompt encoding and action dependency modeling are valid.
- Presentation of the experiment results are comprehensive, and extensive details are given for the method explanation.
- Experimentation is rigorous and follows prior benchmarking.

**Weaknesses:**

1. The pretraining method is not general enough: it only concern about instruction of "follow motion for ..." for a particular motion trajectory, and therefore it mainly tackles the tasks with prompts given a certain motion of a certain trajectory. This means it assumes the task at hand is always similar to follow motion, which is not true.

- An example to illustrate this: it can do well for task T10, but for task T13, when it sweeps something without touching an object, it cannot generalize.

2. For the pretraining method to work, this method also assumes that the prompts contains the motion trajectory keypoint, which is a very narrow assumption and might not always hold. The end users would not be expected to provide the entire trajectories all the time. Therefore the pretraining on motion following is a bit overfitting to the tasks that VIMA designed.

- related to this point: the work advocates for inverse dynamics modeling, but I think this is quite specific to the VIMA-bench task setting with algorithmic oracle. It would be hard to model inverse dynamics in real world.

3. Effectiveness of proposed method: in terms of experiment results (for the full results in Appendix A), there are not significant improvement over VIMA; for those tasks (T10) that has significant improvement, it seems to because the pretraining phase overfits to "Follow Motion" task.

**Questions:**

1. In Appendix  A page 14, two variations of "Ours" are:
- w/ pretrain
- w/ Encoder-Decoder
Is "w/ Encoder-Decoder" with or without pretraining? Are these two variations adding "pretrain" and "Encoder-Decoder" on top of some common method or is one of them adding on top of another?

3. For task T13, could you provide more details on the failure cases of VIMA and this method respectively? Providing some video rollouts of the two methods would be great.

4. The authors are encouraged to provide full experiment results in the main text rather than a portion of it.

5. The conclusion mentioned that the work "demonstrate the in-context learning capability". Could the authors elaborate more on this?

---

> ### Author Response · Authors · 2023-11-16
> **Response to Reviewer Y3nY**
>
> We appreciate the detailed feedback from the reviewer. Please find our responses below.
>
> > The pretraining method is not general enough: it only concern about instruction of "follow motion for ..." for a particular motion trajectory, and therefore it mainly tackles the tasks with prompts given a certain motion of a certain trajectory.
> >
>
> We would like to clarify an important misunderstanding: The goal of our pretraining is not to achieve zero-shot generalization for any unseen task. Instead, we aim to endow robots with an intuitive, human-like ability to learn and adapt to new tasks through in-context demonstrations. This focus is the cornerstone of our inverse dynamic pretraining design, facilitating the robot's understanding of the underlying transition dynamics suggested by the multimodal prompts.
>
> Nevertheless, our pretraining can still improve our model's performance on tasks without trajectory instructions in the prompt. Evidence of this can be found in Table 1, which shows that our pretraining can improve the performance of Task 5 (Rearrange then restore) consistently on L1, L2, and L3 levels, despite Task 5 only containing subgoal images in its prompt, demonstrating the versatility of our pretraining approach.
>
> > This means it assumes the task at hand is always similar to follow motion.... it can do well for task T10, but for task T13... it cannot generalize.
>
> We contend that the challenges in solving Task 13 (Sweep without Touching) are not indicative of shortcomings in our algorithm design. And thus, tackling Task 13 is outside the scope of our paper.
>
> Task 13 is an L4 task in the VIMA-BENCH, which is out of the training distribution. Our analysis of the VIMA-BENCH training tasks and trajectories reveals two primary factors hindering generalization to Task 13:
>
> 1. **End Effector Utilization**: T13 employs a "spatula" end effector, a tool only used in Task T12 (Sweep without Exceeding) among all training tasks. 1. The majority (12 out of 13) of training tasks utilize a "suction cup" end effector. This distinction is critical as the suction cup is associated with "pick and place" motor skills, while the spatula is linked to "wipe" actions, as detailed in Section 2.
> 2. **Layout and Prompt Specificity**: The workspace layouts of Task 12 and Task 13 are nearly identical, yet they diverge significantly from the other training tasks. Task 13's prompt only differs from T12's by the term touching. However, the training data lacks the necessary signal to effectively conceptualize and respond to touching, impacting the model's performance on Task 13.
>
> Consequently, policies trained via our methods and VIMA tend to replicate Task 12's action sequence in Task 13, leading to failure.
>
> However, we kindly ask the reviewer to pay attention to our method's success in significantly improving performance on Task 5, 9, and 17. This improvement is not limited to motion-following tasks. Furthermore, while Task 10 (Follow Motion) shares similarities with our pretraining prompts, the demonstration image sequence in Task 10 includes distractors absent from the robot's workspace (see Figure 2). Conversely, our pretraining tasks incorporate the robot's observational sequence directly into the task prompt, as illustrated in Figure 4.
>
> > For the pretraining method to work, this method also assumes that the prompts contains the motion trajectory keypoint, which is a very narrow assumption and might not always hold.
>
> First, We would like to clarify the misunderstanding that our pretraining method does not depend on the motion trajectory key point. Our method only requires a sequence of state-action pairs of the robot trajectory, which is always satisfied in the multi-task imitation settings. Second, the key points are a natural consequence of our action space $\mathcal{A} = (\mathcal{T}\_{intial}, \mathcal{T}\_{target})$, as each observation is gathered at the start/end of each action. Moreover, we emphasize that this versatile action space has been widely adopted in various robotics system, including VIMA, Transporter Network[1], and CLIPort [2].
>
> > The end users would not be expected to provide the entire trajectories all the time. Therefore the pretraining on motion following is a bit overfitting to the tasks that VIMA designed.
>
> We are considering multi-task imitation learning where demonstration trajectories for each training task are provided. We argue that multi-task imitation learning is a general setting adopted in various works [1, 2]. We can always conduct our pretraining for any robot trajectory with a sequence state-action of pair [3]. Therefore, our pretraining strategies are NOT overfitting to the VIMA designed.

---

> > ### Author Response · Authors · 2023-11-16
> > **(continued) Response to Reviewer Y3nY**
> >
> > > It would be hard to model inverse dynamics in real world
> >
> > We argue that it is possible to perform inverse dynamics in real-world scenarios [4, 5, 6]. Radosavovic et al. perform masked sensorimotor pretraining by training a model to predict the masked-out trajectory from the rest [4]. Note that `Our Method w/ masked pretrain` can be seen as a special case of it. However, we find that casting the inverse dynamic prediction as a motion-following pretraining task (Sec. 3.1) benefits the most when tackling the multimodal prompt in our case.
> >
> > > there are not significant improvement over VIMA; for those tasks (T10) that has significant improvement, it seems to because the pretraining phase overfits to "Follow Motion" task.
> >
> > Again, we emphasize that there are significant differences between our pretraining tasks and Task 10. Task 10's robot trajectory in the prompt differs from the execution time (Figure 2 in paper). In contrast, our pretraining tasks always use the same robot trajectory in the prompt as the execution trajectory. The difference leads to the results in Table 2, where `Our Method w/ Pretrain Only` (without multi-task finetuning) cannot perform well on T10.
> >
> > In terms of the improvement over VIMA, we highlight that our methods improve over VIMA by 86.5% on T9 (Twist), 29.0% on T5 (Rearrange the restore), and 19.5% on T17 (Pick then restore). None of these tasks are within the "one-shot demonstration" category defined by the VIMA-BENCH.
> >
> > > w/ pretrain, w/ Encoder-Decoder Is "w/ Encoder-Decoder"
> >
> > `w/ pretrain` is simply our full methods.
> > `w/ Encoder-Decoder` differs from our full methods by replacing the decoder-only architecture of our policy with the Encoder-Decoder architecture.
> >
> > > For task T13, could you provide more details on the failure cases of VIMA and this method respectively? Providing some video rollouts of the two methods would be great.
> >
> > As suggested by the reviewer, we include the video in the [link](https://imgur.com/a/LGfh5cm). Both our policy and VIMA policy learn spurious correlation and approach T13 the same as T12.
> >
> > > The authors are encouraged to provide full experiment results in the main text rather than a portion of it.
> >
> > We thank the reviewer for the suggestion. Due to the space limit, we did not include the full experiment results in the main text. We followed a similar approach to the VIMA paper, which also presents overall success rates for each evaluation level and defers detailed experiment results to the Appendix. We are willing to incorporate the full experiment results into the main text should our paper be accepted and are granted an additional page to accommodate this expansion.
> >
> > > The conclusion mentioned that the work "demonstrate the in-context learning capability". Could the authors elaborate more on this?
> >
> > In terms of in-context learning ability, we aim to endow robots with an intuitive, human-like ability to learn and adapt to new tasks through in-context demonstrations. A comprehensive description and a series of experiments dedicated to evaluating our policy's in-context learning are detailed in Section 4.3 of our paper. To further substantiate our claims, we designed four new testing tasks, each accompanied by in-context examples in their prompts. Appendix D provided detailed task definitions and evaluation results. We kindly suggest the reviewer revisit these sections.
> >
> > [1] Zeng et al., Transporter Networks: Rearranging the Visual World for Robotic Manipulation, CoRL 2020.
> >
> > [2] Shridhar et al., CLIPORT: What and Where Pathways for Robotic Manipulation, CoRL 2021.
> >
> > [3] Reed, Scott, et al., A generalist agent.  TMLR
> >
> > [4] Radosavovic et al., Robot Learning with Sensorimotor Pre-training, arXiv 2023.
> >
> > [5] Nguyen-Tuong, et al. "Learning inverse dynamics: a comparison." *European symposium on artificial neural networks*. No. CONF. 2008.
> >
> > [6] Hitzler, et al. "Learning and adaptation of inverse dynamics models: A comparison."  *2019 IEEE-RAS 19th International Conference on Humanoid Robots (Humanoids)*. IEEE, 2019.
> >
> > [7] Valencia-Vidal et al. "Bidirectional recurrent learning of inverse dynamic models for robots with elastic joints: a real-time real-world implementation." *Frontiers in Neurorobotics* 17 (2023): 1166911.

---

> ### Author Response · Authors · 2023-11-21
> **Follow up the discussion**
>
> Dear Reviewer Y3nY,
>
> Thank you again for your time and effort. Your feedback has been valuable in helping us clarify, improve, and refine our work. We have carefully addressed your comments in our authors' responses to improve the quality of our paper. We thus kindly request that you take a moment to revisit our paper and consider the changes we have made. We hope our clarifications warrant a more positive evaluation of our work.
>
> Best regards,
>
> The authors

---

> > ### Comment · Reviewer_Y3nY · 2023-11-21
> > **Thank you for your response!**
> >
> > I thank the authors for their responses and I acknowledge the clarifications by the authors. Thank you and the comments will be taken into considerations.

---

> > > ### Author Response · Authors · 2023-11-22
> > > **Thank you!**
> > >
> > > Dear Reviewer Y3nY,
> > >
> > > We appreciate your willingness to reconsider the rating of our work! We are readily available to provide clarification or further information if you have additional questions or concerns!
> > >
> > > Best regards,
> > >
> > > The Authors

---

> > > ### Author Response · Authors · 2023-11-23
> > > **Inquiry about the evaluation**
> > >
> > > Dear Reviewer Y3nY,
> > >
> > > As the discussion period is coming to an end, we wonder if we have addressed your concerns and answered your questions. If yes, would you kindly consider raising the score? Thanks again for your very constructive and thoughtful comments!
> > >
> > > Best regards,
> > >
> > > The Authors

---

### Official Review · Reviewer_YMJx · 2023-11-01

**Soundness:** 3 good
**Presentation:** 3 good
**Contribution:** 2 fair
**Rating:** 5
**Confidence:** 4

**Summary:**

This paper studies the problem of multi-modal prompting in "embodied tasks", i.e., the combination of language and image to train a model to be capable of multi-tasks. The authors introduced a two-stage training pipeline, in pretraining, using the inverse dynamic modelling loss, and in fine-tuning, using a multi-task imitation loss.

Overall, this paper can be seen as a follow-up of the VIMA[1] paper. Results show a 10% success rate gain in the VIMA benchmark.

**Strengths:**

The paper is well written. I am glad to read the detailed analysis of the ablation studies. The introduction and related works sections indicate the authors are very familiar with relevant literature.

**Weaknesses:**

While this paper looks technically sound to me, I found the small improvements based on the VIMA paper can not be viewed as a significant contribution that is sufficient to be accepted in ICLR. The claimed contributions include (1) a MIDAS training framework, i.e., introducing inverse dynamic modelling loss, page 5 Eq(3) in pretraining + multi-task imitation loss; (2) residual connections in the visual layers;
(3) a small performance gain (10%) compared to the VIMA paper. However, using inverse dynamic modelling loss and multi-task supervision loss are all intuitive and an easy follow-up step after the VIMA paper. Therefore, the reviewer found the contributions are not sufficient to be published as a long paper in ICLR.

Nov 23 update: regarding (3), after reviewing the additional experiments the authors submitted, I think the performance looks good for me. I will raise my score to 5 accordingly.

**Questions:**

- The authors add the appendix pages in the main paper, which exceeds the page limits. Please remove the appendix in the revision.
Nov 23 update: Have no concerns after rebuttal.

---

> ### Author Response · Authors · 2023-11-16
> **Response to Reviewer YMJx**
>
> We would like to clarify several major misunderstandings on the contributions of our work.
>
> > using inverse dynamic modeling loss and multi-task supervision loss are all intuitive and an easy follow-up step after the VIMA paper
>
> We would like to emphasize that our core novelty does not lie in the specific training losses employed. It lies in equipping a multi-task policy with the capacity for in-context learning, which is achieved by applying a two-stage training pipeline. Our goal is to endow a multi-task robot with an intuitive, human-like ability to learn and adapt to new tasks through in-context demonstrations rather than striving for zero-shot generalization to entirely unseen tasks. Table 1, 2, and Appendix D shows that our method establishes SOTA performances across all four evaluation protocols while exhibiting a superior in-context learning ability. To the best of our knowledge, **simultaneously equipping a robot with multi-task and in-context learning abilities** has not been extensively explored in robotics, making our contribution a significant step forward in this domain.
>
> > (2) residual connections in the visual layers
>
> We would like to clarify that our contribution is **an effective multimodal prompt encoder that can capture visual and textual details**. Specifically, it is constructed by adding a residual connection (RC) from the input visual tokens to the encoded embeddings, assisting the encoding process to retain more detailed visual information. Notably, this simple yet effective design is not limited to the task of robotics control. It can be incorporated into any multimodal LLM (MLLM), e.g., LLaVA [1] and AnyMAL [2],  to facilitate general multimodal understanding.
>
> > (3) a small performance gain (10%) compared to the VIMA paper
>
> Our method achieves average success rates of 97.8% (+10.6% over VIMA) on L1, 97.9% (+10.9% over VIMA) on L2,  93.4% (+9.4% over VIMA) on L3,  and 59.1% (+9.5% over VIMA) on L4 on the standard VIMA-BENCH. Notably, we highlight our exceptional gains on Task 5 (Rearrange the restore, +31.8%), Task 9 (Twist, +86.3%), Task 10 (Follow Motion, + 41.0%) and  Task17 (Pick then restore, +19.3%). These results clearly indicate a major advancement over the VIMA model, not just a small performance gain.
>
> > The reviewer found the contributions are not sufficient to be published as a long paper in ICLR.
>
> We kindly remind that the reviewer has overlooked our policy's superior in-context learning ability, as empirically supported in Section 4.3 and Appendix D. Our model can learn from in-context examples for novel tasks, a feature where the baseline VIMA model falls short. We highlight the in-context learning ability is crucial for a generalist robot.
>
> We assert the novelty of our research. Prior works have primarily focused on developing either a multi-task robot or one capable of learning from demonstrations or in-context examples. However, integrating in-context learning into a multi-task policy remains under-explored in robotics.
>
> Moreover, our work sheds light on the emergence of the in-context learning ability in our model, a topic of significant interest in NLP. While one might initially credit inverse dynamic pretraining for this ability, Table 2 reveals that policies developed through our two-stage pipeline (both `Our Method` and `Our Method w/o Modified FT`) outperform those from inverse dynamic pretraining alone (`Our Method w/ Pretrain Only`). This phenomenon echoes the findings in FLAN [3], which indicate that a finetuned language model excels in in-context learning.
>
> We thank the reviewer's invaluable feedback, which helped us clarify our contributions. We kindly ask the reviewer to revisit our paper in light of our response and consider whether the clarifications we have provided might warrant a reconsideration of the rating.
>
> > The authors add the appendix pages in the main paper, which exceeds the page limits. Please remove the appendix in revision.
>
> We follow the official instructions of ICLR https://iclr.cc/Conferences/2024/CallForPapers. We make sure our main text is within the 9 pages limit.
> **Paper length**
>
> There will be a strict upper limit of 9 pages for the main text of the submission, with unlimited additional pages for citations. This page limit applies to both the initial and final camera ready version.
>
> - Authors may use as many pages of appendices (after the bibliography) as they wish, but reviewers are not required to read the appendix.
>
> [1] Liu et al., Visual Instruction Tuning. NeurIPS 2023
>
> [2] Moon et al., AnyMAL: An Efficient and Scalable Any-Modality Augmented Language Model. arXiv:2309.16058
>
> [3] Wei et al. "Finetuned language models are zero-shot learners." ICLR 2022

---

> ### Author Response · Authors · 2023-11-21
> **Follow up the discussion**
>
> Dear Reviewer YMJx,
>
> Thank you again for your time and effort. Your feedback has been valuable in helping us clarify, improve, and refine our work. We have carefully addressed your comments in our authors' responses to improve the quality of our paper. We thus kindly request that you take a moment to revisit our paper and consider the changes we have made. We hope our clarifications warrant a more positive evaluation of our work.
>
> Best regards,
>
> The authors

---

> > ### Author Response · Authors · 2023-11-22
> > **Looking forward to your response!**
> >
> > Dear Reviewer YMJx,
> >
> > Thank you again for your feedback! We have meticulously addressed each of your concerns in our response and revised manuscript. We look forward to your response and hope that our clarifications will contribute to an improved evaluation of our paper.
> >
> > Best regards,
> >
> > The authors

---

### Official Review · Reviewer_t7wu · 2023-11-09

**Soundness:** 3 good
**Presentation:** 3 good
**Contribution:** 3 good
**Rating:** 6
**Confidence:** 3

**Summary:**

Good paper! This paper proposes a new method called MIDAS for robot manipulation with multimodal prompts. The key ideas are:
A two-stage training pipeline with inverse dynamics pretraining and multi-task finetuning
An effective multimodal prompt encoder that augments a pretrained language model with a residual connection to visual features
Modeling action dimensions as individual tokens and decoding them autoregressively
The method is evaluated on the VIMA-BENCH benchmark and establishes a new state-of-the-art, improving success rate by around 10%. The ablation studies demonstrate the benefits of the proposed training strategy and prompt encoder design.

**Strengths:**

- The inverse dynamics pretraining is an interesting idea to enable the model to infer actions from visual observations. This facilitates in-context learning from demonstration examples in the prompts.

- Modeling action dimensions independently and decoding them autoregressively is intuitive and shows improved performance.

- Comprehensive experiments on the challenging VIMA-BENCH benchmark with clear improvements over prior state-of-the-art.

- Ablation studies provide useful insights into the contribution of different components.

**Weaknesses:**

The prompts are quite controlled during pretraining versus the more complex prompts at test time. It is unclear if the pretraining fully transfers to the downstream tasks.

**Questions:**

- For the inverse dynamics pretraining, were other self-supervised objectives explored besides simply reconstructing the actions?

- What stopped the baseline VIMA model from reaching the same performance with just more compute/data?

- Is there other complementary information like force sensors that could augment the visual observations?

---

> ### Author Response · Authors · 2023-11-16
> **Response to Reviewer t7wu**
>
> We are grateful for the reviewer's positive feedback on our work. We hope the following response will further clarify any concerns and assist in advocating for the acceptance of our paper.
>
> > The prompts are quite controlled during pretraining versus the more complex prompts at test time. It is unclear if the pretraining fully transfers to the downstream tasks.
>
> Our pretraining mainly focuses on enabling the agent to reason the inverse dynamics from the task prompt, and thus does not incorporate a wide range of language. Our approach can still benefit the downstream tasks without demonstration trajectory in the prompt. For example, it can improve the performance of Task 5 (Rearrange then restore) consistently across L1, L2, and L3, despite Task 5 only containing subgoal images in the prompt.
>
> We acknowledge the potential benefits of a more varied language in our pretraining prompts and consider its inclusion as a valuable future extension of our research.
>
> > For the inverse dynamics pretraining, were other self-supervised objectives explored besides simply reconstructing the actions?
>
> In this project, we only tried the inverse dynamic prediction as the pretraining tasks given limited resources. However, it might also be possible to reconstruct the observation images during pretraining, as shown by Radosavovic et al. [1]
>
> > What stopped the baseline VIMA model from reaching the same performance with just more compute/data?
>
> 1. The VIMA models each action dimension independently, which can lead to task failure when tackling tasks requiring coordination between the initial and target pose of the action.
> 2. The training data provided in VIMA-BENCH is imbalanced. We reveal this problem by finetuning a VIMA policy trained on the full multi-task data on only the trajectories of Task 5 (Twist). Its success rate increased from ~13% to ~92% on L1, L2, and L3 evaluations.
> 3. Performing multi-task imitation alone is insufficient to enable the VIMA policy to reason inverse dynamics given novel tasks with in-context examples in the prompt, which leads to the VIMA policy's inability to perform, e.g., Task 9 (Twist) and Task 10 (Follow Motion).
>
> > Is there other complementary information like force sensors that could augment the visual observations?
>
> The current VIMA-BENCH only provides visual observations. However, in general, we can always include complementary information to augment the observation space, and this state-based information, like force sensors, can often improve the sample efficiency during training. For example, Radosavovic et al. incorporate proprioceptive robot states to assist in action prediction.
>
> However, in-context examples/demonstrations are more accessible to gather as a sequence of images/video when tackling novel unseen tasks. Therefore, we consider pure image observations in this paper.
>
>
> [1] Radosavovic et al., Robot Learning with Sensorimotor Pre-training, arXiv 2023.

---

> ### Author Response · Authors · 2023-11-21
> **Follow up the discussion**
>
> Dear Reviewer t7wu,
>
> Thank you again for your time and effort. Your feedback has been valuable in helping us clarify, improve, and refine our work. We have carefully addressed your comments in our authors' responses to improve the quality of our paper. We thus kindly request that you take a moment to revisit our paper and consider the changes we have made. We hope our clarifications warrant a more positive evaluation of our work.
>
> Best regards,
>
> The authors

---

> > ### Author Response · Authors · 2023-11-23
> > **Looking forward to your response!**
> >
> > Dear Reviewer t7wu,
> >
> > We appreciate your insightful reviews and positive feedback on our work! As the discussion period is coming to an end, we would like to inquire whether our responses and clarifications address your concerns. We hope the revision we made increases your confidence in recommending the acceptance of our work!
> >
> > We look forward to your response!
> >
> > Best regards,
> >
> > The authors

---

### Author Response · Authors · 2023-11-16
**General response: new tasks!**

We thank all reviewers for their constructive feedback! We have updated our manuscript and responded to individual reviews below. Here, we would like to bring your attention to items that may interest all of you.

1. One of our key contributions is equipping a multi-task policy with the in-context learning ability. To the best of our knowledge, **simultaneously equipping a robot with multi-task and in-context learning abilities** has not been extensively explored in prior robotic research. We have provided detailed experimental results for this in Sec. 4.3 and Appendix D.
2. To strengthen our in-context learning results, we expand the VIMA-BENCH by designing four new tasks, each accompanied by in-context examples in the prompt, as detailed in Appendix D. Evaluation results further showcase our policy's superior in-context learning ability. Notably, these tasks can also facilitate future research towards this line.
3. Our pretraining aims to enhance the robot's understanding of the underlying transition dynamics implied by multimodal prompts. Importantly, our objective is not to achieve zero-shot generalization to entirely novel tasks. Instead, we aim to foster an intuitive, human-like capacity in robots for learning and adapting to new tasks through in-context demonstrations.

---

### Comment · Area_Chair_5a8y · 2023-11-23
**Author-Reviewer discussion period ending *very* soon**

Thank you to the reviewers who responded during the discussion phase. The authors have put great effort into their response, so can I please urge reviewers **t7wu** and **YMJx** to answer the rebuttal.
Thank you!

---

### Meta-Review · Area_Chair_5a8y · 2023-12-06

**Metareview:**

The paper focuses on enhancing robot manipulation through a framework that incorporates multimodal prompts, combining visual cues with textual descriptions. While prompt-based learning has seen success in language tasks for large language models (LLMs), this study addresses the challenge of enabling robots to comprehend and act upon combined vision and language signals. The proposed framework involves a two-stage training process: inverse dynamics pretraining and multi-task finetuning. To facilitate multimodal understanding, they augment a pretrained language model with a connection to visual input and model dependencies among action dimensions.

Reviewers noted several strengths, including the innovative use of inverse dynamics pretraining for action inference from visual prompts, the intuitive modeling of action dimensions, and the comprehensive experiments demonstrating clear improvements over prior benchmarks, notably on the VIMA-BENCH dataset. Additionally, the paper's well-written nature, detailed ablation studies, and extensive understanding of relevant literature were highlighted as positives. However, reviewers raised several concerns. These encompassed doubts regarding the transferability of controlled pretraining prompts to more complex test-time prompts, debates about the significance of the marginal improvement (10%) over previous work for acceptance in the conference, and the narrow focus of the pretraining method on specific tasks that might limit its generalizability. There were also reservations about the feasibility of inverse dynamics modeling in real-world scenarios and the lack of evaluation on broader benchmarks or real-world manipulation demonstrations.

Following the rebuttal, recommendations from reviews sat at 2 marginal rejects (5) and 2 marginal accepts (6). Authors expressed concern over lack of response of review YMJx, who did respond during the AC-Reviewer discussion phase. They noted that they appreciated the additional experiments, and that these were satisfied. However no comment on the novelty/impact concern. The score was raised, but still sat at a marginal reject. In general, there was no strong push to accept this paper from any of the reviewers.

Reviewers suggested potential avenues for improvement, including exploring other benchmarks, addressing generalizability concerns, and demonstrating real-world applicability, while also suggesting additions to citations for recent related work.

**Justification For Why Not Higher Score:**

Reviewers seemed mixed.It is unclear if the issues raised during the rebuttal period were adequately addressed. Ever after reviewers responded, they did not increase their scored beyond a 6.

**Justification For Why Not Lower Score:**

N/A

---

### Decision · Program_Chairs · 2024-01-16

Reject